# DISCRETE VARIATIONAL AUTOENCODERS

**Jason Tyler Rolfe**
D-Wave Systems
Burnaby, BC V5G-4M9, Canada
`jrolfe@dwavesys.com`

## ABSTRACT

Probabilistic models with discrete latent variables naturally capture datasets composed of discrete classes. However, they are difficult to train efficiently, since backpropagation through discrete variables is generally not possible. We present a novel method to train a class of probabilistic models with discrete latent variables using the variational autoencoder framework, including backpropagation through the discrete latent variables. The associated class of probabilistic models comprises an undirected discrete component and a directed hierarchical continuous component. The discrete component captures the distribution over the disconnected smooth manifolds induced by the continuous component. As a result, this class of models efficiently learns both the class of objects in an image, and their specific realization in pixels, from unsupervised data; and outperforms state-of-the-art methods on the permutation-invariant MNIST, Omniglot, and Caltech-101 Silhouettes datasets.

## 1 INTRODUCTION

Unsupervised learning of probabilistic models is a powerful technique, facilitating tasks such as denoising and inpainting, and regularizing supervised tasks such as classification (Hinton et al., 2006; Salakhutdinov & Hinton, 2009; Rasmus et al., 2015). Many datasets of practical interest are projections of underlying distributions over real-world objects into an observation space; the pixels of an image, for example. When the real-world objects are of discrete types subject to continuous transformations, these datasets comprise multiple disconnected smooth manifolds. For instance, natural images change smoothly with respect to the position and pose of objects, as well as scene lighting. At the same time, it is extremely difficult to directly transform the image of a person to one of a car while remaining on the manifold of natural images.

It would be natural to represent the space within each disconnected component with continuous variables, and the selection amongst these components with discrete variables. In contrast, most state-of-the-art probabilistic models use exclusively discrete variables — as do DBMs (Salakhutdinov & Hinton, 2009), NADEs (Larochelle & Murray, 2011), sigmoid belief networks (Spiegelhalter & Lauritzen, 1990; Bornschein et al., 2016), and DARNs (Gregor et al., 2014) — or exclusively continuous variables — as do VAEs (Kingma & Welling, 2014; Rezende et al., 2014) and GANs (Goodfellow et al., 2014).[1] Moreover, it would be desirable to apply the efficient variational autoencoder framework to models with discrete values, but this has proven difficult, since backpropagation through discrete variables is generally not possible (Bengio et al., 2013; Raiko et al., 2015).

We introduce a novel class of probabilistic models, comprising an undirected graphical model defined over binary latent variables, followed by multiple directed layers of continuous latent variables. This class of models captures both the discrete class of the object in an image, and its specific continuously deformable realization. Moreover, we show how these models can be trained efficiently using the variational autoencoder framework, including backpropagation through the binary latent variables. We ensure that the evidence lower bound remains tight by incorporating a hierarchical approximation to the posterior distribution of the latent variables, which can model strong correlations. Since these models efficiently marry the variational autoencoder framework with discrete latent variables, we call them *discrete variational autoencoders* (discrete VAEs).

---

[1]Spike-and-slab RBMs (Courville et al., 2011) use both discrete and continuous latent variables.

## 1.1 Variational autoencoders are incompatible with discrete distributions

Conventionally, unsupervised learning algorithms maximize the log-likelihood of an observed dataset under a probabilistic model. Even stochastic approximations to the gradient of the log-likelihood generally require samples from the posterior and prior of the model. However, sampling from undirected graphical models is generally intractable (Long & Servedio, 2010), as is sampling from the posterior of a directed graphical model conditioned on its leaf variables (Dagum & Luby, 1993).

In contrast to the exact log-likelihood, it can be computationally efficient to optimize a lower bound on the log-likelihood (Jordan et al., 1999), such as the evidence lower bound (ELBO, $\mathcal{L}(x, \theta, \phi)$; Hinton & Zemel, 1994):

$$\mathcal{L}(x, \theta, \phi) = \log p(x|\theta) - \text{KL}[q(z|x, \phi)||p(z|x, \theta)], \tag{1}$$

where $q(z|x, \phi)$ is a computationally tractable approximation to the posterior distribution $p(z|x, \theta)$. We denote the observed random variables by $x$, the latent random variables by $z$, the parameters of the generative model by $\theta$, and the parameters of the approximating posterior by $\phi$. The variational autoencoder (VAE; Kingma & Welling, 2014; Rezende et al., 2014; Kingma et al., 2014) regroups the evidence lower bound of Equation 1 as:

$$\mathcal{L}(x, \theta, \phi) = - \underbrace{\text{KL}\left[q(z|x, \phi)||p(z|\theta)\right]}_{\text{KL term}} + \underbrace{\mathbb{E}_q\left[\log p(x|z, \theta)\right]}_{\text{autoencoding term}}. \tag{2}$$

In many cases of practical interest, such as Gaussian $q(z|x)$ and $p(z)$, the KL term of Equation 2 can be computed analytically. Moreover, a low-variance stochastic approximation to the gradient of the autoencoding term can be obtained using backpropagation and the reparameterization trick, so long as samples from the approximating posterior $q(z|x)$ can be drawn using a differentiable, deterministic function $f(x, \phi, \rho)$ of the combination of the inputs, the parameters, and a set of input- and parameter-independent random variables $\rho \sim D$. For instance, samples can be drawn from a Gaussian distribution with mean and variance determined by the input, $\mathcal{N}(m(x, \phi), v(x, \phi))$, using $f(x, \phi, \rho) = m(x, \phi) + \sqrt{v(x, \phi)} \cdot \rho$, where $\rho \sim \mathcal{N}(0, 1)$. When such an $f(x, \phi, \rho)$ exists,

$$\frac{\partial}{\partial \phi} \mathbb{E}_{q(z|x, \phi)}\left[\log p(x|z, \theta)\right] \approx \frac{1}{N} \sum_{\rho \sim \mathcal{D}} \frac{\partial}{\partial \phi} \log p(x|f(x, \rho, \phi), \theta). \tag{3}$$

The reparameterization trick can be generalized to a large set of distributions, including nonfactorial approximating posteriors. We address this issue carefully in Appendix A, where we find that an analog of Equation 3 holds. Specifically, $\mathcal{D}_i$ is the uniform distribution between 0 and 1, and

$$f(x) = \mathbf{F}^{-1}(x), \tag{4}$$

where $\mathbf{F}$ is the conditional-marginal cumulative distribution function (CDF) defined by:

$$F_i(\mathbf{x}) = \int_{x_i'=-\infty}^{x} p\left(x_i'|x_1, \ldots, x_{i-1}\right). \tag{5}$$

However, this generalization is only possible if the inverse of the conditional-marginal CDF exists and is differentiable.

A formulation comparable to Equation 3 is not possible for discrete distributions, such as restricted Boltzmann machines (RBMs) (Smolensky, 1986):

$$p(z) = \frac{1}{\mathcal{Z}_p} e^{-E_p(z)} = \frac{1}{\mathcal{Z}_p} \cdot e^{\left(z^\top W z + b^\top z\right)}, \tag{6}$$

where $z \in \{0, 1\}^n$, $\mathcal{Z}_p$ is the partition function of $p(z)$, and the lateral connection matrix $W$ is triangular. Any approximating posterior that only assigns nonzero probability to a discrete domain corresponds to a CDF that is piecewise-contant. That is, the range of the CDF is a proper subset of the interval $[0, 1]$. The domain of the inverse CDF is thus also a proper subset of $[0, 1]$, and its derivative is not defined, as required in Equations 3 and 4.[2]

---

[2] This problem remains even if we use the quantile function, $F_p^{-1}(\rho) = \inf \left\{ z \in \mathbb{R} : \int_{z'=-\infty}^{z} p(z') \geq \rho \right\}$, the derivative of which is either zero or infinite if $p$ is a discrete distribution.

In the following sections, we present the discrete variational autoencoder (discrete VAE), a hierarchical probabilistic model consising of an RBM,[3] followed by multiple directed layers of continuous latent variables. This model is efficiently trainable using the variational autoencoder formalism, as in Equation 3, including backpropagation through its discrete latent variables.

## 1.2 RELATED WORK

Recently, there have been many efforts to develop effective unsupervised learning techniques by building upon variational autoencoders. Importance weighted autoencoders (Burda et al., 2016), Hamiltonian variational inference (Salimans et al., 2015), normalizing flows (Rezende & Mohamed, 2015), and variational Gaussian processes (Tran et al., 2016) improve the approximation to the posterior distribution. Ladder variational autoencoders (Sønderby et al., 2016) increase the power of the architecture of both approximating posterior and prior. Neural adaptive importance sampling (Du et al., 2015) and reweighted wake-sleep (Bornschein & Bengio, 2015) use sophisticated approximations to the gradient of the log-likelihood that do not admit direct backpropagation. Structured variational autoencoders use conjugate priors to construct powerful approximating posterior distributions (Johnson et al., 2016).

It is easy to construct a stochastic approximation to the gradient of the ELBO that admits both discrete and continuous latent variables, and only requires computationally tractable samples. Unfortunately, this naive estimate is impractically high-variance, leading to slow training and poor performance (Paisley et al., 2012). The variance of the gradient can be reduced somewhat using the baseline technique, originally called REINFORCE in the reinforcement learning literature (Mnih & Gregor, 2014; Williams, 1992; Mnih & Rezende, 2016), which we discuss in greater detail in Appendix B.

Prior efforts by Makhzani et al. (2015) to use multimodal priors with implicit discrete variables governing the modes did not successfully align the modes of the prior with the intrinsic clusters of the dataset. Rectified Gaussian units allow spike-and-slab sparsity in a VAE, but the discrete variables are also implicit, and their prior factorial and thus unimodal (Salimans, 2016). Graves (2016) computes VAE-like gradient approximations for mixture models, but the component models are assumed to be simple factorial distributions. In contrast, discrete VAEs generalize to powerful multimodal priors on the discrete variables, and a wider set of mappings to the continuous units.

The generative model underlying the discrete variational autoencoder resembles a deep belief network (DBN; Hinton et al., 2006). A DBN comprises a sigmoid belief network, the top layer of which is conditioned on the visible units of an RBM. In contrast to a DBN, we use a bipartite Boltzmann machine, with both sides of the bipartite split connected to the rest of the model. Moreover, all hidden layers below the bipartite Boltzmann machine are composed of continuous latent variables with a fully autoregressive layer-wise connection architecture. Each layer $j$ receives connections from all previous layers $i < j$, with connections from the bipartite Boltzmann machine mediated by a set of smoothing variables. However, these architectural differences are secondary to those in the gradient estimation technique. Whereas DBNs are traditionally trained by unrolling a succession of RBMs, discrete variational autoencoders use the reparameterization trick to backpropagate through the evidence lower bound.

## 2 BACKPROPAGATING THROUGH DISCRETE LATENT VARIABLES BY ADDING CONTINUOUS LATENT VARIABLES

When working with an approximating posterior over discrete latent variables, we can effectively smooth the conditional-marginal CDF (defined by Equation 5 and Appendix A) by augmenting the latent representation with a set of continous random variables. The conditional-marginal CDF over the new continuous variables is invertible and its inverse is differentiable, as required in Equations 3 and 4. We redefine the generative model so that the conditional distribution of the observed variables given the latent variables only depends on the new continuous latent space. This does not alter

---

[3]Strictly speaking, the prior contains a bipartite Boltzmann machine, all the units of which are connected to the rest of the model. In contrast to a traditional RBM, there is no distinction between the "visible" units and the "hidden" units. Nevertheless, we use the familiar term RBM in the sequel, rather than the more cumbersome "fully hidden bipartite Boltzmann machine."

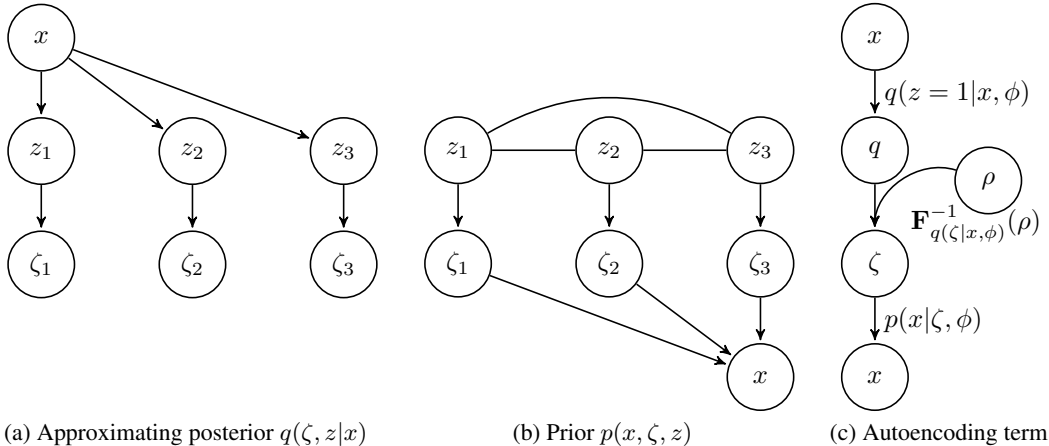

(a) Approximating posterior $q(\zeta, z|x)$ (b) Prior $p(x, \zeta, z)$ (c) Autoencoding term

Figure 1: Graphical models of the smoothed approximating posterior (a) and prior (b), and the network realizing the autoencoding term of the ELBO from Equation 2 (c). Continuous latent variables $\zeta_i$ are smoothed analogs of discrete latent variables $z_i$, and insulate $z$ from the observed variables $x$ in the prior (b). This facilitates the marginalization of the discrete $z$ in the autoencoding term of the ELBO, resulting in a network (c) in which all operations are deterministic and differentiable given independent stochastic input $\rho \sim U[0, 1]$.

the fundamental form of the model, or the KL term of Equation 2; rather, it can be interpreted as adding a noisy nonlinearity, like dropout (Srivastava et al., 2014) or batch normalization with a small minibatch (Ioffe & Szegedy, 2015), to each latent variable in the approximating posterior and the prior. The conceptual motivation for this approach is discussed in Appendix C.

Specifically, as shown in Figure 1a, we augment the latent representation in the approximating posterior with continuous random variables $\zeta$,[4] conditioned on the discrete latent variables $z$ of the RBM:

$$q(\zeta, z|x, \phi) = r(\zeta|z) \cdot q(z|x, \phi), \qquad \text{where}$$
$$r(\zeta|z) = \prod_i r(\zeta_i|z_i).$$

The support of $r(\zeta|z)$ for all values of $z$ must be connected, so the marginal distribution $q(\zeta|x, \phi) = \sum_z r(\zeta|z) \cdot q(z|x, \phi)$ has a constant, connected support so long as $0 < q(z|x, \phi) < 1$. We further require that $r(\zeta|z)$ is continuous and differentiable except at the endpoints of its support, so the inverse conditional-marginal CDF of $q(\zeta|x, \phi)$ is differentiable in Equations 3 and 4, as we discuss in Appendix A.

As shown in Figure 1b, we correspondingly augment the prior with $\zeta$:

$$p(\zeta, z|\theta) = r(\zeta|z) \cdot p(z|\theta),$$

where $r(\zeta|z)$ is the same as for the approximating posterior. Finally, we require that the conditional distribution over the observed variables only depends on $\zeta$:

$$p(x|\zeta, z, \theta) = p(x|\zeta, \theta). \tag{7}$$

The smoothing distribution $r(\zeta|z)$ transforms the model into a continuous function of the distribution over $z$, and allows us to use Equations 2 and 3 directly to obtain low-variance stochastic approximations to the gradient.

Given this expansion, we can simplify Equations 3 and 4 by dropping the dependence on $z$ and applying Equation 16 of Appendix A, which generalizes Equation 3:

$$\frac{\partial}{\partial \phi} \mathbb{E}_{q(\zeta, z|x, \phi)} \left[ \log p(x|\zeta, z, \theta) \right] \approx \frac{1}{N} \sum_{\rho \sim U(0,1)^n} \frac{\partial}{\partial \phi} \log p\left( x | \mathbf{F}^{-1}_{q(\zeta|x, \phi)}(\rho), \theta \right). \tag{8}$$

---

[4]We always use a variant of $z$ for latent variables. This is zeta, or Greek $z$. The discrete latent variables $z$ can conveniently be thought of as English $z$.

If the approximating posterior is factorial, then each $F_i$ is an independent CDF, without conditioning or marginalization.

As we shall demonstrate in Section 2.1, $\mathbf{F}^{-1}_{q(\zeta|x,\phi)}(\rho)$ is a function of $q(z = 1|x, \phi)$, where $q(z = 1|x, \phi)$ is a deterministic probability value calculated by a parameterized function, such as a neural network. The autoencoder implicit in Equation 8 is shown in Figure 1c. Initially, input $x$ is passed into a deterministic feedforward network $q(z = 1|x, \phi)$, for which the final nonlinearity is the logistic function. Its output $q$, along with an independent random variable $\rho \sim U[0, 1]$, is passed into the deterministic function $\mathbf{F}^{-1}_{q(\zeta|x,\phi)}(\rho)$ to produce a sample of $\zeta$. This $\zeta$, along with the original input $x$, is finally passed to $\log p(x|\zeta, \theta)$. The expectation of this log probability with respect to $\rho$ is the autoencoding term of the VAE formalism, as in Equation 2. Moreover, conditioned on the input and the independent $\rho$, this autoencoder is deterministic and differentiable, so backpropagation can be used to produce a low-variance, computationally-efficient approximation to the gradient.

## 2.1 SPIKE-AND-EXPONENTIAL SMOOTHING TRANSFORMATION

As a concrete example consistent with sparse coding, consider the spike-and-exponential transformation from binary $z$ to continuous $\zeta$:

$$r(\zeta_i|z_i = 0) = \begin{cases} \infty, & \text{if } \zeta_i = 0 \\ 0, & \text{otherwise} \end{cases} \qquad F_{r(\zeta_i|z_i=0)}(\zeta') = 1$$

$$r(\zeta_i|z_i = 1) = \begin{cases} \frac{\beta e^{\beta\zeta}}{e^\beta - 1}, & \text{if } 0 \le \zeta_i \le 1 \\ 0, & \text{otherwise} \end{cases} \qquad F_{r(\zeta_i|z_i=1)}(\zeta') = \left.\frac{e^{\beta\zeta}}{e^\beta - 1}\right|_0^{\zeta'} = \frac{e^{\beta\zeta'} - 1}{e^\beta - 1}$$

where $F_p(\zeta') = \int_{-\infty}^{\zeta'} p(\zeta) \cdot d\zeta$ is the CDF of probability distribution $p$ in the domain $[0, 1]$. This transformation from $z_i$ to $\zeta_i$ is invertible: $\zeta_i = 0 \Leftrightarrow z_i = 0$, and $\zeta_i > 0 \Leftrightarrow z_i = 1$ almost surely.[5]

We can now find the CDF for $q(\zeta|x, \phi)$ as a function of $q(z = 1|x, \phi)$ in the domain $[0, 1]$, marginalizing out the discrete $z$:

$$F_{q(\zeta|x,\phi)}(\zeta') = (1 - q(z = 1|x, \phi)) \cdot F_{r(\zeta_i|z_i=0)}(\zeta') + q(z = 1|x, \phi) \cdot F_{r(\zeta_i|z_i=1)}(\zeta')$$

$$= q(z = 1|x, \phi) \cdot \left(\frac{e^{\beta\zeta'} - 1}{e^\beta - 1} - 1\right) + 1.$$

To evaluate the autoencoder of Figure 1c, and through it the gradient approximation of Equation 8, we must invert the conditional-marginal CDF $F_{q(\zeta|x,\phi)}$:

$$F^{-1}_{q(\zeta|x,\phi)}(\rho) = \begin{cases} \frac{1}{\beta} \cdot \log\left[\left(\frac{\rho+q-1}{q}\right) \cdot \left(e^\beta - 1\right) + 1\right], & \text{if } \rho \ge 1 - q \\ 0, & \text{otherwise} \end{cases} \tag{9}$$

where we use the substitution $q(z = 1|x, \phi) \to q$ to simplify notation. For all values of the independent random variable $\rho \sim U[0, 1]$, the function $F^{-1}_{q(\zeta|x,\phi)}(\rho)$ rectifies the input $q(z = 1|x, \phi)$ if $q \le 1 - \rho$ in a manner analogous to a rectified linear unit (ReLU), as shown in Figure 2a. It is also quasi-sigmoidal, in that $F^{-1}$ is increasing but concave-down if $q > 1 - \rho$. The effect of $\rho$ on $F^{-1}$ is qualitatively similar to that of dropout (Srivastava et al., 2014), depicted in Figure 2b, or the noise injected by batch normalization (Ioffe & Szegedy, 2015) using small minibatches, shown in Figure 2c.

Other expansions to the continuous space are possible. In Appendix D.1, we consider the case where both $r(\zeta_i|z_i = 0)$ and $r(\zeta_i|z_i = 1)$ are linear functions of $\zeta$; in Appendix D.2, we develop a spike-and-slab transformation; and in Appendix E, we explore a spike-and-Gaussian transformation where the continuous $\zeta$ is directly dependent on the input $x$ in addition to the discrete $z$.

---

[5]In the limit $\beta \to \infty$, $\zeta_i = z_i$ almost surely, and the continuous variables $\zeta$ can effectively be removed from the model. This trick can be used after training with finite $\beta$ to produce a model without smoothing variables $\zeta$.

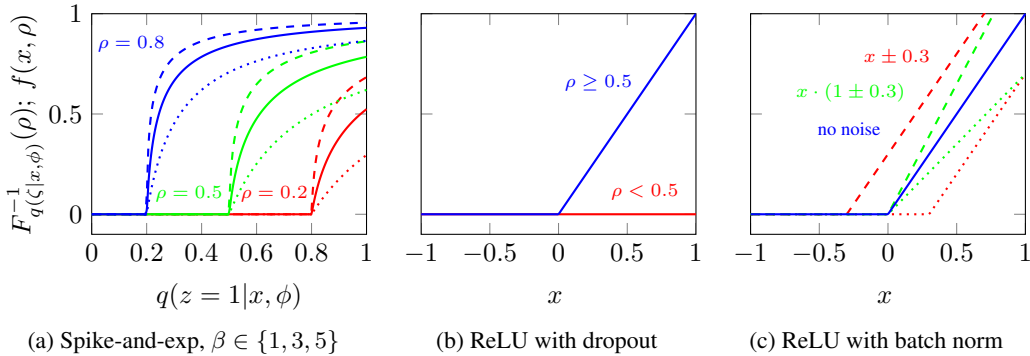

(a) Spike-and-exp, $\beta \in \{1, 3, 5\}$   (b) ReLU with dropout   (c) ReLU with batch norm

Figure 2: Inverse CDF of the spike-and-exponential smoothing transformation for $\rho \in \{0.2, 0.5, 0.8\}$; $\beta = 1$ (dotted), $\beta = 3$ (solid), and $\beta = 5$ (dashed) (a). Rectified linear unit with dropout rate 0.5 (b). Shift (red) and scale (green) noise from batch normalization; with magnitude 0.3 (dashed), $-0.3$ (dotted), or 0 (solid blue); before a rectified linear unit (c). In all cases, the abcissa is the input and the ordinate is the output of the effective transfer function. The novel stochastic nonlinearity $F_{q(\zeta|x,\phi)}^{-1}(\rho)$ from Figure 1c, of which (a) is an example, is qualitatively similar to the familiar stochastic nonlinearities induced by dropout (b) or batch normalization (c).

## 3 ACCOMMODATING EXPLAINING-AWAY WITH A HIERARCHICAL APPROXIMATING POSTERIOR

When a probabilistic model is defined in terms of a prior distribution $p(z)$ and a conditional distribution $p(x|z)$, the observation of $x$ often induces strong correlations in the posterior $p(z|x)$ due to phenomena such as explaining-away (Pearl, 1988). Moreover, we wish to use an RBM as the prior distribution (Equation 6), which itself may have strong correlations. In contrast, to maintain tractability, many variational approximations use a product of independent approximating posterior distributions (e.g., mean-field methods, but also Kingma & Welling (2014); Rezende et al. (2014)).

To accommodate strong correlations in the posterior distribution while maintaining tractability, we introduce a hierarchy into the approximating posterior $q(z|x)$ over the discrete latent variables. Specifically, we divide the latent variables $z$ of the RBM into disjoint groups, $z_1, \ldots, z_k$,[6] and define the approximating posterior via a directed acyclic graphical model over these groups:

$$q(z_1, \zeta_1, \ldots, z_k, \zeta_k | x, \phi) = \prod_{1 \le j \le k} r(\zeta_j | z_j) \cdot q(z_j | \zeta_{i<j}, x, \phi) \qquad \text{where}$$

$$q(z_j | \zeta_{i<j}, x, \phi) = \frac{e^{g_j(\zeta_{i<j}, x, \phi)^\top \cdot z_j}}{\prod_{z_\iota \in z_j} \left(1 + e^{g_{z_\iota}(\zeta_{i<j}, x, \phi)}\right)}, \tag{10}$$

$z_j \in \{0, 1\}^n$, and $g_j(\zeta_{i<j}, x, \phi)$ is a parameterized function of the inputs and preceding $\zeta_i$, such as a neural network. The corresponding graphical model is depicted in Figure 3a, and the integration of such hierarchical approximating posteriors into the reparameterization trick is discussed in Appendix A. If each group $z_j$ contains a single variable, this dependence structure is analogous to that of a deep autoregressive network (DARN; Gregor et al., 2014), and can represent any distribution. However, the dependence of $z_j$ on the preceding discrete variables $z_{i<j}$ is always mediated by the continuous variables $\zeta_{i<j}$.

This hierarchical approximating posterior does not affect the form of the autoencoding term in Equation 8, except to increase the depth of the autoencoder, as shown in Figure 3b. The deterministic probability value $q(z_j = 1 | \zeta_{i<j}, x, \phi)$ of Equation 10 is parameterized, generally by a neural network, in a manner analogous to Section 2. However, the final logistic function is made explicit in Equation 10 to simplify Equation 12. For each successive layer $j$ of the autoencoder, input $x$ and all previous $\zeta_{i<j}$ are passed into the network computing $q(z = 1 | \zeta_{i<j}, x, \phi)$. Its output $q_j$, along with an

---

[6]The continuous latent variables $\zeta$ are divided into complementary disjoint groups $\zeta_1, \ldots, \zeta_k$.

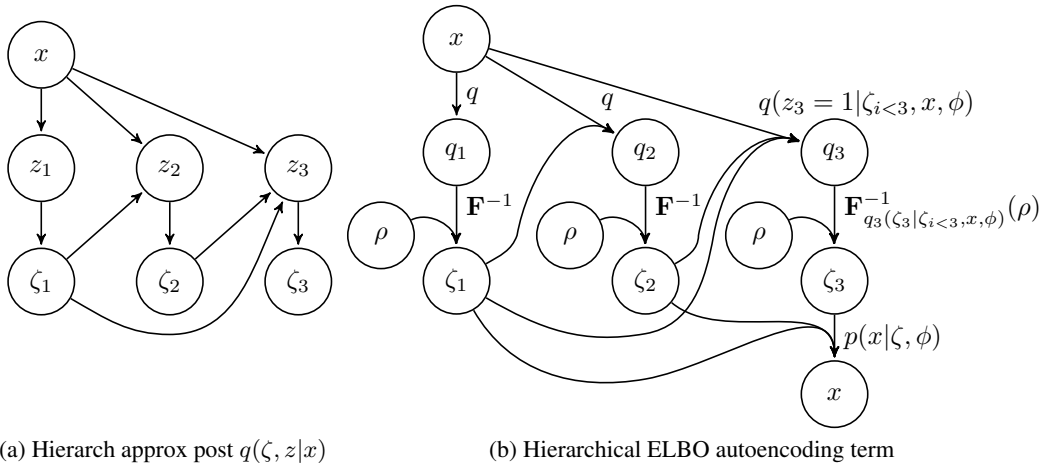

(a) Hierarch approx post $q(\zeta, z|x)$ (b) Hierarchical ELBO autoencoding term

Figure 3: Graphical model of the hierarchical approximating posterior (a) and the network realizing the autoencoding term of the ELBO (b) from Equation 2. Discrete latent variables $z_j$ only depend on the previous $z_{i<j}$ through their smoothed analogs $\zeta_{i<j}$. The autoregressive hierarchy allows the approximating posterior to capture correlations and multiple modes. Again, all operations in (b) are deterministic and differentiable given the stochastic input $\rho$.

independent random variable $\rho \sim U[0, 1]$, is passed to the deterministic function $\mathbf{F}^{-1}_{q(\zeta_j|\zeta_{i<j},x,\phi)}(\rho)$ to produce a sample of $\zeta_j$. Once all $\zeta_j$ have been recursively computed, the full $\zeta$ along with the original input $x$ is finally passed to $\log p(x|\zeta, \theta)$. The expectation of this log probability with respect to $\rho$ is again the autoencoding term of the VAE formalism, as in Equation 2.

In Appendix F, we show that the gradients of the remaining KL term of the ELBO (Equation 2) can be estimated stochastically using:

$$\frac{\partial}{\partial\theta}\mathrm{KL}\left[q||p\right] = \mathbb{E}_{q(z_1|x,\phi)}\left[\cdots\left[\mathbb{E}_{q(z_k|\zeta_{i<k},x,\phi)}\left[\frac{\partial E_p(z,\theta)}{\partial\theta}\right]\right]\right] - \mathbb{E}_{p(z|\theta)}\left[\frac{\partial E_p(z,\theta)}{\partial\theta}\right] \quad \text{and}$$

(11)

$$\frac{\partial}{\partial\phi}\mathrm{KL}\left[q||p\right] = \mathbb{E}_\rho\left[(g(x,\zeta) - b)^\top \cdot \frac{\partial q}{\partial\phi} - z^\top \cdot W \cdot \left(\frac{1-z}{1-q} \odot \frac{\partial q}{\partial\phi}\right)\right]. \tag{12}$$

In particular, Equation 12 is substantially lower variance than the naive approach to calculate $\frac{\partial}{\partial\phi}\mathrm{KL}\left[q||p\right]$, based upon REINFORCE.

## 4 MODELLING CONTINUOUS DEFORMATIONS WITH A HIERARCHY OF CONTINUOUS LATENT VARIABLES

We can make both the generative model and the approximating posterior more powerful by adding additional layers of latent variables below the RBM. While these layers can be discrete, we focus on continuous variables, which have proven to be powerful in generative adversarial networks (Goodfellow et al., 2014) and traditional variational autoencoders (Kingma & Welling, 2014; Rezende et al., 2014). When positioned below and conditioned on a layer of discrete variables, continuous variables can build continuous manifolds, from which the discrete variables can choose. This complements the structure of the natural world, where a percept is determined first by a discrete selection of the types of objects present in the scene, and then by the position, pose, and other continuous attributes of these objects.

Specifically, we augment the latent representation with continuous random variables $\mathfrak{z}$,[7] and define both the approximating posterior and the prior to be layer-wise fully autoregressive directed graphical models. We use the same autoregressive variable order for the approximating posterior as for the

---

[7]We always use a variant of $z$ for latent variables. This is Fraktur $z$, or German $z$.

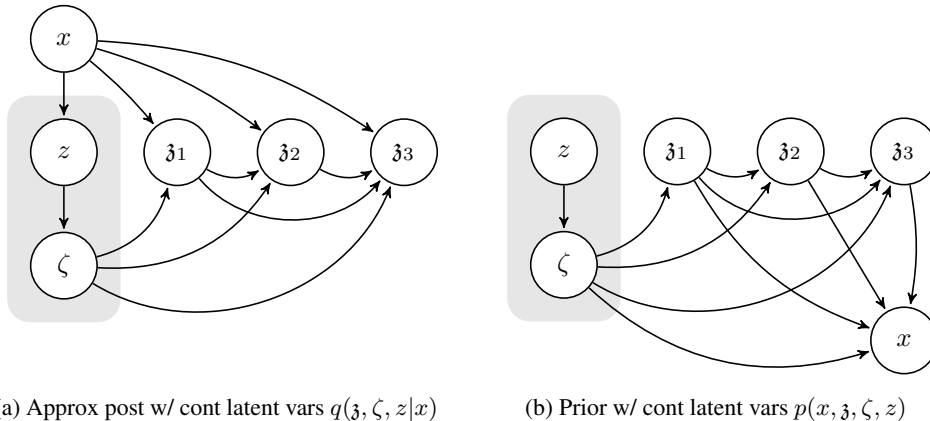

(a) Approx post w/ cont latent vars $q(\mathfrak{z}, \zeta, z|x)$ (b) Prior w/ cont latent vars $p(x, \mathfrak{z}, \zeta, z)$

Figure 4: Graphical models of the approximating posterior (a) and prior (b) with a hierarchy of continuous latent variables. The shaded regions in parts (a) and (b) expand to Figures 3a and 1b respectively. The continuous latent variables $\mathfrak{z}$ build continuous manifolds, capturing properties like position and pose, conditioned on the discrete latent variables $z$, which can represent the discrete types of objects in the image.

prior, as in DRAW (Gregor et al., 2015), variational recurrent neural networks (Chung et al., 2015), the deep VAE of Salimans (2016), and ladder networks (Rasmus et al., 2015; Sønderby et al., 2016). We discuss the motivation for this ordering in Appendix G.

The directed graphical model of the approximating posterior and prior are defined by:

$$q(\mathfrak{z}_0, \ldots, \mathfrak{z}_n | x, \phi) = \prod_{0 \le m \le n} q\left(\mathfrak{z}_m | \mathfrak{z}_{l<m}, x, \phi\right) \qquad \text{and}$$

$$p(\mathfrak{z}_0, \ldots, \mathfrak{z}_n | \theta) = \prod_{0 \le m \le n} p\left(\mathfrak{z}_m | \mathfrak{z}_{l<m}, \theta\right). \tag{13}$$

The full set of latent variables associated with the RBM is now denoted by $\mathfrak{z}_0 = \{z_1, \zeta_1, \ldots, z_k, \zeta_k\}$. However, the conditional distributions in Equation 13 only depend on the continuous $\zeta_j$. Each $\mathfrak{z}_{m \ge 1}$ denotes a layer of continuous latent variables, and Figure 4 shows the resulting graphical model.

The ELBO decomposes as:

$$\mathcal{L}(x, \theta, \phi) = \mathbb{E}_{q(\mathfrak{z}|x,\phi)} \left[\log p(x|\mathfrak{z}, \theta)\right] - \sum_m \mathbb{E}_{q(\mathfrak{z}_{l<m}|x,\phi)} \left[\mathrm{KL}\left[q(\mathfrak{z}_m | \mathfrak{z}_{l<m}, x, \phi) || p(\mathfrak{z}_m | \mathfrak{z}_{l<m}, \theta)\right]\right].$$

$$\tag{14}$$

If both $q(\mathfrak{z}_m | \mathfrak{z}_{l<m}, x, \phi)$ and $p(\mathfrak{z}_m | \mathfrak{z}_{l<m}, \theta)$ are Gaussian, then their KL divergence has a simple closed form, which is computationally efficient if the covariance matrices are diagonal. Gradients can be passed through the $q(\mathfrak{z}_{l<m} | x, \phi)$ using the traditional reparameterization trick, described in Section 1.1.

## 5 RESULTS

Discrete variational autoencoders comprise a smoothed RBM (Section 2) with a hierarchical approximating posterior (Section 3), followed by a hierarchy of continuous latent variables (Section 4). We parameterize all distributions with neural networks, except the smoothing distribution $r(\zeta|z)$ discussed in Section 2. Like NVIL (Mnih & Gregor, 2014) and VAEs (Kingma & Welling, 2014; Rezende et al., 2014), we define all approximating posteriors $q$ to be explicit functions of $x$, with parameters $\phi$ shared between all inputs $x$. For distributions over discrete variables, the neural networks output the parameters of a factorial Bernoulli distribution using a logistic final layer, as in Equation 10; for the continuous $\mathfrak{z}$, the neural networks output the mean and log-standard deviation of a diagonal-covariance Gaussian distribution using a linear final layer. Each layer of the neural networks parameterizing the distributions over $z$, $\mathfrak{z}$, and $x$ consists of a linear transformation,

batch normalization (Ioffe & Szegedy, 2015) (but see Appendix H.2), and a rectified-linear point-wise nonlinearity (ReLU). We stochastically approximate the expectation with respect to the RBM prior $p(z|\theta)$ in Equation 11 using block Gibbs sampling on persistent Markov chains, analogous to persistent contrastive divergence (Tieleman, 2008). We minimize the ELBO using ADAM (Kingma & Ba, 2015) with a decaying step size.

The hierarchical structure of Section 4 is very powerful, and overfits without strong regularization of the prior, as shown in Appendix H. In contrast, powerful approximating posteriors do not induce significant overfitting. To address this problem, we use conditional distributions over the input $p(x|\zeta, \theta)$ without any deterministic hidden layers, except on Omniglot. Moreover, all other neural networks in the prior have only one hidden layer, the size of which is carefully controlled. On statically binarized MNIST, Omniglot, and Caltech-101, we share parameters between the layers of the hierarchy over $\mathfrak{z}$. We present the details of the architecture in Appendix H.

We train the resulting discrete VAEs on the permutation-invariant MNIST (LeCun et al., 1998), Omniglot[8] (Lake et al., 2013), and Caltech-101 Silhouettes datasets (Marlin et al., 2010). For MNIST, we use both the static binarization of Salakhutdinov & Murray (2008) and dynamic binarization. Estimates of the log-likelihood[9] of these models, computed using the method of (Burda et al., 2016) with $10^4$ importance-weighted samples, are listed in Table 1. The reported log-likelihoods for discrete VAEs are the average of 16 runs; the standard deviation of these log-likelihoods are 0.08, 0.04, 0.05, and 0.11 for dynamically and statically binarized MNIST, Omniglot, and Caltech-101 Silhouettes, respectively. Removing the RBM reduces the test set log-likelihood by 0.09, 0.37, 0.69, and 0.66.

| MNIST (dynamic binarization) | | MNIST (static binarization) | | |
|---|---|---|---|---|
| | **LL** | | **ELBO** | **LL** |
| DBN | -84.55 | HVI | -88.30 | -85.51 |
| IWAE | -82.90 | DRAW | -87.40 | |
| Ladder VAE | -81.74 | NAIS NADE | | -83.67 |
| Discrete VAE | **-80.15** | Normalizing flows | -85.10 | |
| | | Variational Gaussian process | | -81.32 |
| | | Discrete VAE | **-84.58** | **-81.01** |
| Omniglot | | Caltech-101 Silhouettes | | |
| | **LL** | | | **LL** |
| IWAE | -103.38 | IWAE | | -117.2 |
| Ladder VAE | -102.11 | RWS SBN | | -113.3 |
| RBM | -100.46 | RBM | | -107.8 |
| DBN | -100.45 | NAIS NADE | | -100.0 |
| Discrete VAE | **-97.43** | Discrete VAE | | **-97.6** |

Table 1: Test set log-likelihood of various models on the permutation-invariant MNIST, Omniglot, and Caltech-101 Silhouettes datasets. For the discrete VAE, the reported log-likelihood is estimated with $10^4$ importance-weighted samples (Burda et al., 2016). For comparison, we also report performance of some recent state-of-the-art techniques. Full names and references are listed in Appendix I.

We further analyze the performance of discrete VAEs on dynamically binarized MNIST: the largest of the datasets, requiring the least regularization. Figure 5 shows the generative output of a discrete VAE as the Markov chain over the RBM evolves via block Gibbs sampling. The RBM is held constant across each sub-row of five samples, and variation amongst these samples is due to the layers of continuous latent variables. Given a multimodal distribution with well-separated modes, Gibbs sampling passes through the large, low-probability space between the modes only infrequently. As a result, consistency of the digit class over many successive rows in Figure 5 indicates that the RBM prior has well-separated modes. The RBM learns distinct, separated modes corresponding to the different digit types, except for 3/5 and 4/9, which are either nearby or overlapping; at least tens of

---

[8]We use the partitioned, preprocessed Omniglot dataset of Burda et al. (2016), available from https://github.com/yburda/iwae/tree/master/datasets/OMNIGLOT.

[9]The importance-weighted estimate of the log-likelihood is a lower bound, except for the log partition function of the RBM. We describe our unbiased estimation method for the partition function in Appendix H.1.

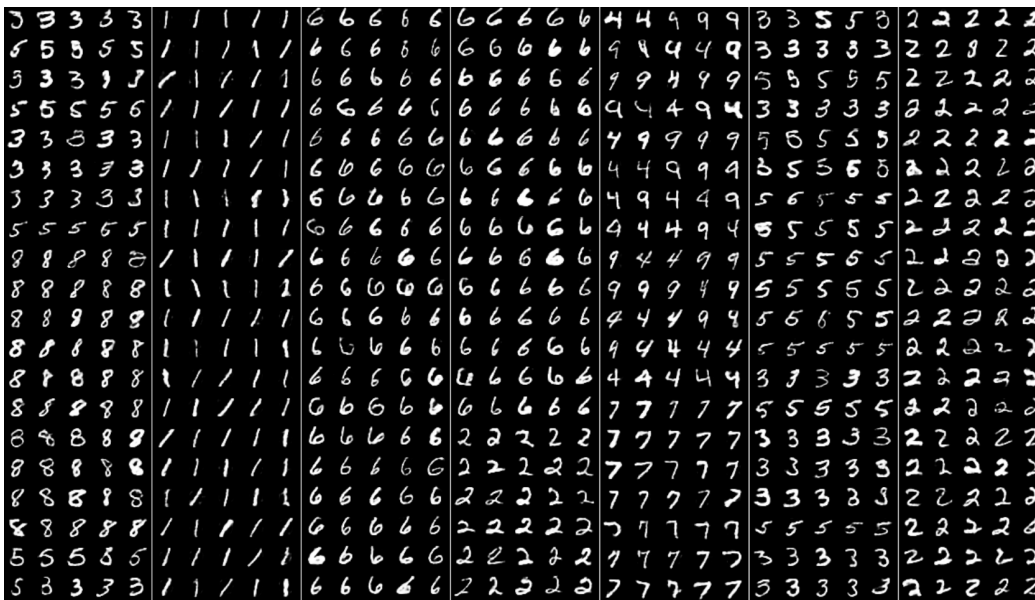

Figure 5: Evolution of samples from a discrete VAE trained on dynamically binarized MNIST, using persistent RBM Markov chains. We perform 100 iterations of block-Gibbs sampling on the RBM between successive rows. Each horizontal group of 5 uses a single, shared sample from the RBM, but independent continuous latent variables, and shows the variation induced by the continuous layers as opposed to the RBM. The long vertical sequences in which the digit ID remains constant demonstrate that the RBM has well-separated modes, each of which corresponds to a single (or occasionally two) digit IDs, despite being trained in a wholly unsupervised manner.

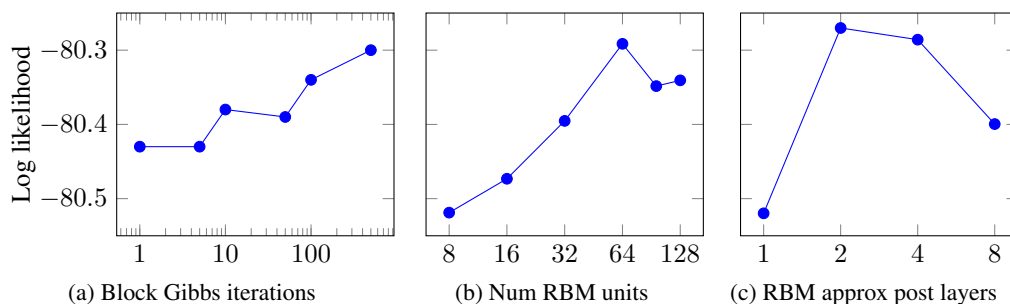

| (a) Block Gibbs iterations | (b) Num RBM units | (c) RBM approx post layers |

Figure 6: Log likelihood versus the number of iterations of block Gibbs sampling per minibatch (a), the number of units in the RBM (b), and the number of layers in the approximating posterior over the RBM (c). Better sampling (a) and hierarchical approximating posteriors (c) support better performance, but the network is robust to the size of the RBM (b).

thousands of iterations of single-temperature block Gibbs sampling is required to mix between the modes. We present corresponding figures for the other datasets, and results on simplified architectures, in Appendix J.

The large mixing time of block Gibbs sampling on the RBM suggests that training may be constrained by sample quality. Figure 6a shows that performance[10] improves as we increase the number of iterations of block Gibbs sampling performed per minibatch on the RBM prior: $p(z|\theta)$ in Equation 11. This suggests that a further improvement may be achieved by using a more effective sampling algorithm, such as parallel tempering (Swendsen & Wang, 1986).

---

[10]All models in Figure 6 use only 10 layers of continuous latent variables, for computational efficiency.

Commensurate with the small number of intrinsic classes, a moderately sized RBM yields the best performance on MNIST. As shown in Figure 6b, the log-likelihood plateaus once the number of units in the RBM reaches at least 64. Presumably, we would need a much larger RBM to model a dataset like Imagenet, which has many classes and complicated relationships between the elements of various classes.

The benefit of the hierarchical approximating posterior over the RBM, introduced in Section 3, is apparent from Figure 6c. The reduction in performance when moving from 4 to 8 layers in the approximating posterior may be due to the fact that each additional hierarchical layer over the approximating posterior adds three layers to the encoder neural network: there are two deterministic hidden layers for each stochastic latent layer. As a result, expanding the number of RBM approximating posterior layers significantly increases the number of parameters that must be trained, and increases the risk of overfitting.

## 6 CONCLUSION

Datasets consisting of a discrete set of classes are naturally modeled using discrete latent variables. However, it is difficult to train probabilistic models over discrete latent variables using efficient gradient approximations based upon backpropagation, such as variational autoencoders, since it is generally not possible to backpropagate through a discrete variable (Bengio et al., 2013).

We avoid this problem by symmetrically projecting the approximating posterior and the prior into a continuous space. We then evaluate the autoencoding term of the evidence lower bound exclusively in the continous space, marginalizing out the original discrete latent representation. At the same time, we evaluate the KL divergence between the approximating posterior and the true prior in the original discrete space; due to the symmetry of the projection into the continuous space, it does not contribute to the KL term. To increase representational power, we make the approximating posterior over the discrete latent variables hierarchical, and add a hierarchy of continuous latent variables below them. The resulting discrete variational autoencoder achieves state-of-the-art performance on the permutation-invariant MNIST, Omniglot, and Caltech-101 Silhouettes datasets.

ACKNOWLEDGEMENTS

Zhengbing Bian, Fabian Chudak, Arash Vahdat helped run experiments. Jack Raymond provided the library used to estimate the log partition function of RBMs. Mani Ranjbar wrote the cluster management system, and a custom GPU acceleration library used for an earlier version of the code. We thank Evgeny Andriyash, William Macready, and Aaron Courville for helpful discussions; and one of our anonymous reviewers for identifying the problem addressed in Appendix D.3.

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

## A    MULTIVARIATE VAES BASED ON THE CUMULATIVE DISTRIBUTION FUNCTION

The reparameterization trick is always possible if the cumulative distribution function (CDF) of $q(z|x, \phi)$ is invertible, and the inverse CDF is differentiable, as noted in Kingma & Welling (2014). However, for multivariate distributions, the CDF is defined by:

$$F(\mathbf{x}) = \int_{x_1' = -\infty}^{x_1} \cdots \int_{x_n' = -\infty}^{x_n} p(x_1', \ldots, x_n').$$

The multivariate CDF maps $\mathcal{R}^n \to [0, 1]$, and is generally *not* invertible.[11]

In place of the multivariate CDF, consider the set of conditional-marginal CDFs defined by:[12]

$$F_i(\mathbf{x}) = \int_{x'_i = -\infty}^{x} p\left(x'_i | x_1, \ldots, x_{i-1}\right). \tag{15}$$

That is, $F_j(\mathbf{x})$ is the CDF of $x_j$, conditioned on all $x_i$ such that $i < h$, and marginalized over all $x_k$ such the $j < k$. The range of each $F_j$ is $[0, 1]$, so $\mathbf{F}$ maps the domain of the original distribution to $\rho \in [0, 1]^n$. To invert $\mathbf{F}$, we need only invert each conditional-marginal CDF in turn, conditioning $x_j = F_j^{-1}(\rho)$ on $x_1 = F_1^{-1}(\rho), \ldots, x_{j-1} = F_{j-1}^{-1}(\rho)$. These inverses exist so long as the conditional-marginal probabilities are everywhere nonzero. It is not problematic to effectively define $F_j^{-1}(\rho)$ based upon $x_{i<j}$, rather than $\rho_{i<j}$, since by induction we can uniquely determine $x_{i<j}$ given $\rho_{i<j}$.

Using integration-by-substition, we can compute the gradient of the ELBO by taking the expectation of a uniform random variable $\rho$ on $[0, 1]^n$, and using $\mathbf{F}_{q(z|x,\phi)}^{-1}$ to transform $\rho$ back to the element of $z$ on which $p(x|z, \theta)$ is conditioned. To perform integration-by-substitution, we will require the determinant of the Jacobian of $\mathbf{F}^{-1}$.

The derivative of a CDF is the probability density function at the selected point, and $F_j$ is a simple CDF when we hold fixed the variables $x_{i<j}$ on which it is conditioned, so using the inverse function theorem we find:

$$\begin{aligned}
\frac{\partial F_j^{-1}(\rho)}{\partial \rho_j} &= \frac{1}{F_j'(F_j^{-1}(\rho))} \\
&= \frac{1}{p\left(x_j = F_j^{-1}(\rho) | x_{i<j},\right)}
\end{aligned}$$

where $\rho$ is a vector, and $F_j'$ is $\frac{\partial F_j}{\partial \rho_j}$. The Jacobian matrix $\frac{\partial \mathbf{F}}{\partial \mathbf{x}}$ is triangular, since the earlier conditional-marginal CDFs $F_j$ are independent of the value of the later $x_k$, $j < k$, over which they are marginalized. Moreover, the inverse conditional-marginal CDFs have the same dependence structure as $\mathbf{F}$, so the Jacobian of $\mathbf{F}^{-1}$ is also triangular. The determinant of a triangular matrix is the product of the diagonal elements.

---

[11]For instance, for the bivariate uniform distribution on the interval $[0, 1]^2$, the CDF $F(x, y) = x \cdot y$ for $0 \le x, y \le 1$, so for any $0 \le c \le 1$ and $c \le x \le 1$, $y = \frac{c}{x}$ yields $F(x, y) = c$. Clearly, many different pairs $(x, y)$ yield each possible value $c$ of $F(x, y)$.

[12]The set of marginal CDFs, used to define copulas, is invertible. However, it does not generally map the original distribution to a simple joint distribution, such as a multivariate unform distribution, as required for variational autoencoders. In Equation 16, $\left|\det\left(\frac{\partial \mathbf{F}_{q(z|x,\phi)}^{-1}(\rho)}{\partial \phi}\right)\right|$ does not cancel out $q\left(\mathbf{F}_{q(z|x,\phi)}^{-1}(\rho) | x, \phi\right)$. The determinant of the inverse Jacobian is instead $\left[\prod_i q\left(z_i = F_i^{-1}(\rho)\right)\right]^{-1}$, which differs from $\left[q\left(\mathbf{F}_{q(z|x,\phi)}^{-1}(\rho)\right)\right]^{-1}$ if $q$ is not factorial. As a result, we do not recover the variational autoencoder formulation of Equation 16.

Using these facts to perform a multivariate integration-by-substitution, we obtain:

$$
\begin{aligned}
\mathbb{E}_{q(z|x,\phi)}\left[\log p(x|z,\theta)\right] &= \int_z q(z|x,\phi) \cdot \log p(x|z,\theta) \\
&= \int_{\rho=\mathbf{0}}^{\mathbf{1}} q\left(\mathbf{F}_{q(z|x,\phi)}^{-1}(\rho)|x,\phi\right) \cdot \log p\left(x|\mathbf{F}_{q(z|x,\phi)}^{-1}(\rho),\theta\right) \cdot \left|\det\left(\frac{\partial \mathbf{F}_{q(z|x,\phi)}^{-1}(\rho)}{\partial \rho}\right)\right| \\
&= \int_{\rho=\mathbf{0}}^{\mathbf{1}} q\left(\mathbf{F}_{q(z|x,\phi)}^{-1}(\rho)|x,\phi\right) \cdot \log p\left(x|\mathbf{F}_{q(z|x,\phi)}^{-1}(\rho),\theta\right) \cdot \left(\prod_j \frac{\partial \mathbf{F}_{q_j(z_j|x,\phi)}^{-1}(\rho_j)}{\partial \rho_j}\right) \\
&= \int_{\rho=\mathbf{0}}^{\mathbf{1}} \frac{q\left(\mathbf{F}_{q(z|x,\phi)}^{-1}(\rho)|x,\phi\right)}{\prod_j q\left(z_j = F_j^{-1}(\rho)|z_{i<j}\right)} \cdot \log p\left(x|\mathbf{F}_{q(z|x,\phi)}^{-1}(\rho),\theta\right) \\
&= \int_{\rho=\mathbf{0}}^{\mathbf{1}} \log p\left(x|\mathbf{F}_{q(z|x,\phi)}^{-1}(\rho),\theta\right) \qquad (16)
\end{aligned}
$$

The variable $\rho$ has dimensionality equal to that of $z$; $\mathbf{0}$ is the vector of all 0s; $\mathbf{1}$ is the vector of all 1s.

The gradient with respect to $\phi$ is then easy to approximate stochastically:

$$
\frac{\partial}{\partial \phi}\mathbb{E}_{q(z|x,\phi)}\left[\log p(x|z,\theta)\right] \approx \frac{1}{N}\sum_{\rho \sim U(0,1)^n} \frac{\partial}{\partial \phi}\log p\left(x|\mathbf{F}_{q(z|x,\phi)}^{-1}(\rho),\theta\right) \qquad (17)
$$

Note that if $q(z|x,\phi)$ is factorial (i.e., the product of independent distributions in each dimension $z_j$), then the conditional-marginal CDFs $F_j$ are just the marginal CDFs in each direction. However, even if $q(z|x,\phi)$ is not factorial, Equation 17 still holds so long as $\mathbf{F}$ is nevertheless defined to be the set of conditional-marginal CDFs of Equation 15.

## B  THE DIFFICULTY OF ESTIMATING GRADIENTS OF THE ELBO WITH REINFORCE

It is easy to construct a stochastic approximation to the gradient of the ELBO that only requires computationally tractable samples, and admits both discrete and continuous latent variables. Unfortunately, this naive estimate is impractically high-variance, leading to slow training and poor performance (Paisley et al., 2012). The variance of the gradient can be reduced somewhat using the baseline technique, originally called REINFORCE in the reinforcement learning literature (Mnih & Gregor, 2014; Williams, 1992; Bengio et al., 2013; Mnih & Rezende, 2016):

$$
\begin{aligned}
\frac{\partial}{\partial \phi}\mathbb{E}_{q(z|x,\phi)}\left[\log p(x|z,\theta)\right] &= \mathbb{E}_{q(z|x,\phi)}\left[\left[\log p(x|z,\theta) - B(x)\right] \cdot \frac{\partial}{\partial \phi}\log q(z|x,\phi)\right] \\
&\approx \frac{1}{N}\sum_{z \sim q(z|x,\phi)}\left(\left[\log p(x|z,\theta) - B(x)\right] \cdot \frac{\partial}{\partial \phi}\log q(z|x,\phi)\right) \qquad (18)
\end{aligned}
$$

where $B(x)$ is a (possibly input-dependent) baseline, which does not affect the gradient, but can reduce the variance of a stochastic estimate of the expectation.

In REINFORCE, $\frac{\partial}{\partial \phi}\mathbb{E}_{q(z|x,\phi)}\left[\log p(x|z,\theta)\right]$ is effectively estimated by something akin to a finite difference approximation to the derivative. The autoencoding term is a function of the conditional log-likelihood $\log p(x|z,\theta)$, composed with the approximating posterior $q(z|x,\phi)$, which determines the value of $z$ at which $p(x|z,\theta)$ is evaluated. However, the conditional log-likelihood is never differentiated directly in REINFORCE, even in the context of the chain rule. Rather, the conditional log-likelihood is evaluated at many different points $z \sim q(z|x,\phi)$, and a weighted sum of these values is used to approximate the gradient, just like in the finite difference approximation.

Equation 18 of REINFORCE captures much less information about $p(x|z,\theta)$ per sample than Equation 3 of the variational autoencoder, which actively makes use of the gradient. In particular, the change of $p(x|z,\theta)$ in some direction $\vec{d}$ can only affect the REINFORCE gradient estimate if a sample is taken with a component in direction $\vec{d}$. In a $D$-dimensional latent space, at least $D$ samples are

required to capture the variation of $p(x|z, \theta)$ in all directions; fewer samples span a smaller subspace. Since the latent representation commonly consists of dozens of variables, the REINFORCE gradient estimate can be much less efficient than one that makes direct use of the gradient of $p(x|z, \theta)$. Moreover, we will show in Section 5 that, when the gradient is calculated efficiently, hundreds of latent variables can be used effectively.

## C  AUGMENTING DISCRETE LATENT VARIABLES WITH CONTINUOUS LATENT VARIABLES

Intuitively, variational autoencoders break the encoder[13] distribution into "packets" of probability of infinitessimal but equal mass, within which the value of the latent variables is approximately constant. These packets correspond to a region $r_i < \rho_i < r_i + \delta$ for all $i$ in Equation 16, and the expectation is taken over these packets. There are more packets in regions of high probability, so high-probability values are more likely to be selected. More rigorously, $\mathbf{F}_{q(z|x,\phi)}(\zeta)$ maps intervals of high probability to larger spans of $0 \leq \rho \leq 1$, so a randomly selected $\rho \sim U[0, 1]$ is more likely to be mapped to a high-probability point by $\mathbf{F}_{q(z|x,\phi)}^{-1}(\rho)$.

As the parameters of the encoder are changed, the location of a packet can move, while its mass is held constant. That is, $\zeta = \mathbf{F}_{q(z|x,\phi)}^{-1}(\rho)$ is a function of $\phi$, whereas the probability mass associated with a region of $\rho$-space is constant by definition. So long as $\mathbf{F}_{q(z|x,\phi)}^{-1}$ exists and is differentiable, a small change in $\phi$ will correspond to a small change in the location of each packet. This allows us to use the gradient of the decoder to estimate the change in the loss function, since the gradient of the decoder captures the effect of small changes in the location of a selected packet in the latent space.

In contrast, REINFORCE (Equation 18) breaks the latent representation into segments of infinitessimal but equal volume; e.g., $z_i \leq z_i' < z_i + \delta$ for all $i$ (Williams, 1992; Mnih & Gregor, 2014; Bengio et al., 2013). The latent variables are also approximately constant within these segments, but the probability mass varies between them. Specifically, the probability mass of the segment $z \leq z' < z + \delta$ is proportional to $q(z|x, \phi)$.

Once a segment is selected in the latent space, its location is independent of the encoder and decoder. In particular, the gradient of the loss function does not depend on the gradient of the decoder with respect to position in the latent space, since this position is fixed. Only the probability mass assigned to the segment is relevant.

Although variational autoencoders can make use of the additional gradient information from the decoder, the gradient estimate is only low-variance so long as the motion of most probability packets has a similar effect on the loss. This is likely to be the case if the packets are tightly clustered (e.g., the encoder produces a Gaussian with low variance, or the spike-and-exponential distribution of Section 2.1), or if the movements of far-separated packets have a similar effect on the total loss (e.g., the decoder is roughly linear).

Nevertheless, Equation 17 of the VAE can be understood in analogy to dropout (Srivastava et al., 2014) or standout (Ba & Frey, 2013) regularization. Like dropout and standout, $\mathbf{F}_{q(z|x,\phi)}^{-1}(\rho)$ is an element-wise stochastic nonlinearity applied to a hidden layer. Since $\mathbf{F}_{q(z|x,\phi)}^{-1}(\rho)$ selects a point in the probability distribution, it rarely selects an improbable point. Like standout, the distribution of the hidden layer is learned. Indeed, we recover the encoder of standout if we use the spike-and-Gaussian distribution of Section E.1 and let the standard deviation $\sigma$ go to zero.

However, variational autoencoders cannot be used directly with discrete latent representations, since changing the parameters of a discrete encoder can only move probability mass between the allowed discrete values, which are far apart. If we follow a probability packet as we change the encoder parameters, it either remains in place, or jumps a large distance. As a result, the vast majority of probability packets are unaffected by small changes to the parameters of the encoder. Even if we are lucky enough to select a packet that jumps between the discrete values of the latent representation,

---

[13]Since the approximating posterior $q(z|x, \phi)$ maps each input to a distribution over the latent space, it is sometimes called the encoder. Correspondingly, since the conditional likelihood $p(x|z, \theta)$ maps each configuration of the latent variables to a distribution over the input space, it is called the decoder.

the gradient of the decoder cannot be used to accurately estimate the change in the loss function, since the gradient only captures the effect of very small movements of the probability packet.

To use discrete latent representations in the variational autoencoder framework, we must first transform to a continuous latent space, within which probability packets move smoothly. That is, we must compute Equation 17 over a different distribution than the original posterior distribution. Surprisingly, we need not sacrifice the original discrete latent space, with its associated approximating posterior. Rather, we extend the encoder $q(z|x, \phi)$ and the prior $p(z|\theta)$ with a transformation to a continuous, auxiliary latent representation $\zeta$, and correspondingly make the decoder a function of this new continuous representation. By extending both the encoder and the prior in the same way, we avoid affecting the remaining KL divergence in Equation 2.[14]

The gradient is defined everywhere if we require that each point in the original latent space map to nonzero probability over the entire auxiliary continuous space. This ensures that, if the probability of some point in the original latent space increases from zero to a nonzero value, no probability packet needs to jump a large distance to cover the resulting new region in the auxiliary continuous space. Moreover, it ensures that the conditional-marginal CDFs are strictly increasing as a function of their main argument, and thus are invertible.

If we ignore the cases where some discrete latent variable has probability 0 or 1, we need only require that, for every pair of points in the original latent space, the associated regions of nonzero probability in the auxiliary continuous space overlap. This ensures that probability packets can move continuously as the parameters $\phi$ of the encoder, $q(z|x, \phi)$, change, redistributing weight amongst the associated regions of the auxiliary continuous space.

## D  ALTERNATIVE TRANSFORMATIONS FROM DISCRETE TO CONTINUOUS LATENT REPRESENTATIONS

The spike-and-exponential transformation from discrete latent variables $z$ to continuous latent variables $\zeta$ presented in Section 2.1 is by no means the only one possible. Here, we develop a collection of alternative transformations.

### D.1  MIXTURE OF RAMPS

As another concrete example, we consider a case where both $r(\zeta_i|z_i = 0)$ and $r(\zeta_i|z_i = 1)$ are linear functions of $\zeta_i$:

$$r(\zeta_i|z_i = 0) = \begin{cases} 2 \cdot (1 - \zeta_i), & \text{if } 0 \leq \zeta_i \leq 1 \\ 0, & \text{otherwise} \end{cases} \qquad F_{r(\zeta_i|z_i=0)}(\zeta') = 2\zeta_i - \zeta_i^2 \big|_0^{\zeta'} = 2\zeta' - \zeta'^2$$

$$r(\zeta_i|z_i = 1) = \begin{cases} 2 \cdot \zeta_i, & \text{if } 0 \leq \zeta_i \leq 1 \\ 0, & \text{otherwise} \end{cases} \qquad F_{r(\zeta_i|z_i=1)}(\zeta') = \zeta_i^2 \big|_0^{\zeta'} = \zeta'^2$$

where $F_p(\zeta') = \int_{-\infty}^{\zeta'} p(\zeta) \cdot d\zeta$ is the CDF of probability distribution $p$ in the domain $[0, 1]$. The CDF for $q(\zeta|x, \phi)$ as a function of $q(z = 1|x, \phi)$ is:

$$\begin{aligned} F_{q(\zeta|x,\phi)}(\zeta') &= (1 - q(z = 1|x, \phi)) \cdot \left(2\zeta' - \zeta'^2\right) + q(z = 1|x, \phi) \cdot \zeta'^2 \\ &= 2 \cdot q(z = 1|x, \phi) \cdot \left(\zeta'^2 - \zeta'\right) + 2\zeta' - \zeta'^2. \end{aligned} \qquad (19)$$

---

[14]Rather than extend the encoder and the prior, we cannot simply prepend the transformation to continuous space to the decoder, since this does not change the space of the probabilty packets.

We can calculate $F^{-1}_{q(\zeta|x,\phi)}$ explicitly, using the substitutions $F_{q(\zeta|x,\phi)} \to \rho$, $q(z = 1|x, \phi) \to q$, and $\zeta' \to \zeta$ in Equation 19 to simplify notation:

$$\rho = 2 \cdot q \cdot (\zeta^2 - \zeta) + 2\zeta - \zeta^2$$
$$0 = (2q - 1) \cdot \zeta^2 + 2(1 - q) \cdot \zeta - \rho$$
$$\zeta = \frac{2(q-1) \pm \sqrt{4(1 - 2q + q^2) + 4(2q - 1)\rho}}{2(2q - 1)}$$
$$= \frac{(q-1) \pm \sqrt{q^2 + 2(\rho - 1)q + (1 - \rho)}}{2q - 1}$$

if $q \neq \frac{1}{2}$; $\rho = \zeta$ otherwise. $F^{-1}_{q(\zeta|x,\phi)}$ has the desired range $[0, 1]$ if we choose

$$F^{-1}(\rho) = \frac{(q-1) + \sqrt{q^2 + 2(\rho - 1)q + (1 - \rho)}}{2q - 1}$$
$$= \frac{q - 1 + \sqrt{(q-1)^2 + (2q - 1) \cdot \rho}}{2q - 1} \tag{20}$$

if $q \neq \frac{1}{2}$, and $F^{-1}(\rho) = \rho$ if $q = \frac{1}{2}$. We plot $F^{-1}_{q(\zeta|x,\phi)}(\rho)$ as a function of $q$ for various values of $\rho$ in Figure 7.

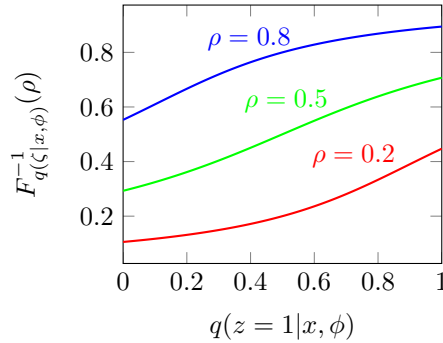

Figure 7: Inverse CDF of the mixture of ramps transformation for $\rho \in \{0.2, 0.5, 0.8\}$

In Equation 20, $F^{-1}_{q(\zeta|x,\phi)}(\rho)$ is quasi-sigmoidal as a function of $q(z = 1|x, \phi)$. If $\rho \ll 0.5$, $F^{-1}$ is concave-up; if $\rho \gg 0.5$, $F^{-1}$ is concave-down; if $\rho \approx 0.5$, $F^{-1}$ is sigmoid. In no case is $F^{-1}$ extremely flat, so it does not kill gradients. In contrast, the sigmoid probability of $z$ inevitably flattens.

## D.2 SPIKE-AND-SLAB

We can also use the spike-and-slab transformation, which is consistent with sparse coding and proven in other successful generative models (Courville et al., 2011):

$$r(\zeta_i|z_i = 0) = \begin{cases} \infty, & \text{if } \zeta_i = 0 \\ 0, & \text{otherwise} \end{cases} \qquad F_{r(\zeta_i|z_i=0)}(\zeta') = 1$$

$$r(\zeta_i|z_i = 1) = \begin{cases} 1, & \text{if } 0 \leq \zeta_i \leq 1 \\ 0, & \text{otherwise} \end{cases} \qquad F_{r(\zeta_i|z_i=1)}(\zeta') = \zeta_i\big|_0^{\zeta'} = \zeta'$$

where $F_p(\zeta') = \int_{-\infty}^{\zeta'} p(\zeta) \cdot d\zeta$ is the cumulative distribution function (CDF) of probability distribution $p$ in the domain $[0, 1]$. The CDF for $q(\zeta|x, \phi)$ as a function of $q(z = 1|x, \phi)$ is:

$$F_{q(\zeta|x,\phi)}(\zeta') = (1 - q(z = 1|x, \phi)) \cdot F_{r(\zeta_i|z_i=0)}(\zeta') + q(z = 1|x, \phi) \cdot F_{r(\zeta_i|z_i=1)}(\zeta')$$
$$= q(z = 1|x, \phi) \cdot (\zeta' - 1) + 1.$$

We can calculate $F^{-1}_{q(\zeta|x,\phi)}$ explicitly, using the substitution $q(z=1|x,\phi) \rightarrow q$ to simplify notation:

$$F^{-1}_{q(\zeta|x,\phi)}(\rho) = \begin{cases} \frac{\rho-1}{q}+1, & \text{if } \rho \geq 1-q \\ 0, & \text{otherwise} \end{cases}$$

We plot $F^{-1}_{q(\zeta|x,\phi)}(\rho)$ as a function of $q$ for various values of $\rho$ in Figure 8.

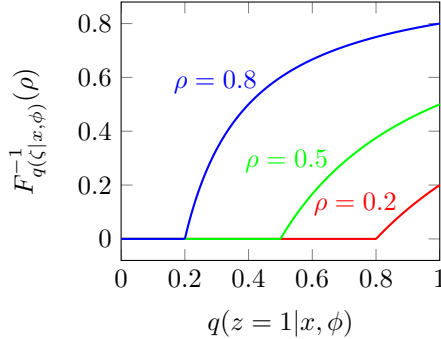

Figure 8: Inverse CDF of the spike-and-slab transformation for $\rho \in \{0.2, 0.5, 0.8\}$

### D.3 ENGINEERING EFFECTIVE SMOOTHING TRANSFORMATIONS

If the smoothing transformation is not chosen appropriately, the contribution of low-probability regions to the expected gradient of the inverse CDF may be large. Using a variant of the inverse function theorem, we find:

$$\frac{\partial}{\partial\theta}F(F^{-1}(\rho)) = \left.\frac{\partial F}{\partial\theta}\right|_{F^{-1}(\rho)} + \left.\frac{\partial F}{\partial z}\right|_{F^{-1}(\rho)} \cdot \frac{\partial}{\partial\theta}F^{-1}(\rho) = 0$$

$$p(z) \cdot \frac{\partial}{\partial\theta}F^{-1}(\rho) = -\left.\frac{\partial F}{\partial\theta}\right|_z, \tag{21}$$

where $z = F^{-1}(\rho)$. Consider the case where $r(\zeta_i|z_i = 0)$ and $r(\zeta_i|z_i = 1)$ are unimodal, but have little overlap. For instance, both distributions might be Gaussian, with means that are many standard deviations apart. For values of $\zeta_i$ between the two modes, $F(\zeta_i) \approx q(z_i = 0|x,\phi)$, assuming without loss of generality that the mode corresponding to $z_i = 0$ occurs at a smaller value of $\zeta_i$ than that corresponding to $z_i = 1$. As a result, $\frac{\partial F}{\partial q} \approx 1$ between the two modes, and $\frac{\partial F^{-1}}{\partial q} \approx \frac{1}{r(\zeta_i)}$ even if $r(\zeta_i) \approx 0$. In this case, the stochastic estimates of the gradient in equation 8, which depend upon $\frac{\partial F^{-1}}{\partial q}$, have large variance.

These high-variance gradient estimates arise because $r(\zeta_i|z_i = 0)$ and $r(\zeta_i|z_i = 1)$ are too well separated, and the resulting smoothing transformation is too sharp. Such disjoint smoothing transformations are analogous to a sigmoid transfer function $\sigma(c \cdot x)$, where $\sigma$ is the logistic function and $c \rightarrow \infty$. The smoothing provided by the continuous random variables $\zeta$ is only effective if there is a region of meaningful overlap between $r(\zeta|z = 0)$ and $r(\zeta|z = 1)$. In particular, $\sum_{z_i} r(\zeta_i|z_i = 0) + r(\zeta_i|z_i = 1) \gg 0$ for all $\zeta_i$ between the modes of $r(\zeta_i|z_i = 0)$ and $r(\zeta_i|z_i = 1)$, so $p(z)$ remains moderate in equation 21. In the spike-and-exponential distribution described in Section 2.1, this overlap can be ensured by fixing or bounding $\beta$.

## E TRANSFORMATIONS FROM DISCRETE TO CONTINUOUS LATENT REPRESENTATIONS THAT DEPEND UPON THE INPUT

It is not necessary to define the transformation from discrete to continuous latent variables in the approximating posterior, $r(\zeta|z)$, to be independent of the input $x$. In the true posterior distribution,

$p(\zeta|z, x) \approx p(\zeta|z)$ only if $z$ already captures most of the information about $x$ and $p(\zeta|z, x)$ changes little as a function of $x$, since

$$p(\zeta|z) = \int_x p(\zeta, x|z) = \int_x p(\zeta|z, x) \cdot p(x|z).$$

This is implausible if the number of discrete latent variables is much smaller than the entropy of the input data distribution. To address this, we can define:

$$q(\zeta, z|x, \phi) = q(z|x, \phi) \cdot q(\zeta|z, x, \phi)$$
$$p(\zeta, z|\theta) = p(\zeta|z) \cdot p(z|\theta)$$

This leads to an evidence lower bound that resembles that of Equation 2, but adds an extra term:

$$
\begin{aligned}
\mathcal{L}_{VAE}(x, \theta, \phi) &= \log p(x|\theta) - \mathrm{KL}\left[q(z, \zeta|x, \phi)||p(z, \zeta|x, \theta)\right] \\
&= \log p(x|\theta) - \mathrm{KL}\left[q(\zeta|z, x, \phi) \cdot q(z|x, \phi)||p(\zeta|z, x, \theta) \cdot p(z|x, \theta)\right] \\
&= \sum_z \int_\zeta q(\zeta|z, x, \phi) \cdot q(z|x, \phi) \cdot \log\left[\frac{p(x|\zeta, \theta) \cdot p(\zeta|z, \theta) \cdot p(z|\theta)}{q(\zeta|z, x, \phi) \cdot q(z|x, \phi)}\right] \\
&= \mathbb{E}_{q(\zeta|z, x, \phi) \cdot q(z|x, \phi)}\left[\log p(x|\zeta, \theta)\right] - \mathrm{KL}\left[q(z|x, \phi)||p(z|\theta)\right] \\
&\quad - \sum_z q(z|x, \phi) \cdot \mathrm{KL}\left[q(\zeta|z, x, \phi)||p(\zeta|z)\right].
\end{aligned}
\tag{22}
$$

The extension to hierarchical approximating posteriors proceeds as in sections 3 and 4.

If both $q(\zeta|z, x, \phi)$ and $p(\zeta|z)$ are Gaussian, then their KL divergence has a simple closed form, which is computationally efficient if the covariance matrices are diagonal. However, while the gradients of this KL divergence are easy to calculate when conditioned on $z$, the gradients with respect of $q(z|x, \phi)$ in the new term seem to force us into a REINFORCE-like approach (c.f. Equation 18):

$$\sum_z \frac{\partial q(z|x, \phi)}{\partial \phi} \cdot \mathrm{KL}\left[q(\zeta|z, x, \phi)||p(\zeta|z)\right] = \mathbb{E}_{q(z|x, \phi)}\left[\mathrm{KL}\left[q(\zeta|z, x, \phi)||p(\zeta|z)\right] \cdot \frac{\partial \log q(z|x, \phi)}{\partial \phi}\right].\tag{23}$$

The reward signal is now $\mathrm{KL}\left[q(\zeta|z, x, \phi)||p(\zeta|z)\right]$ rather than $\log p(x|z, \theta)$, but the effect on the variance is the same, likely negating the advantages of the variational autoencoder in the rest of the loss function.

However, whereas REINFORCE is high-variance because it samples over the expectation, we can perform the expectation in Equation 23 analytically, without injecting any additional variance. Specifically, if $q(z|x, \phi)$ and $q(\zeta|z, x, \phi)$ are factorial, with $q(\zeta_i|z_i, x, \phi)$ only dependent on $z_i$, then $\mathrm{KL}\left[q(\zeta|z, x, \phi)||p(\zeta|z)\right]$ decomposes into a sum of the KL divergences over each variable, as does $\frac{\partial \log q(z|x, \phi)}{\partial \phi}$. The expectation of all terms in the resulting product of sums is zero except those of the form $\mathbb{E}\left[\mathrm{KL}\left[q_i||p_i\right] \cdot \frac{\partial \log q_i}{\partial \phi}\right]$, due to the identity explained in Equation 27. We then use the reparameterization trick to eliminate all hierarchical layers before the current one, and marginalize over each $z_i$. As a result, we can compute the term of Equation 23 by backpropagating

$$\mathrm{KL}\left[q(\zeta|z = 1, x, \phi)||p(\zeta|z = 1)\right] - \mathrm{KL}\left[q(\zeta|z = 0, x, \phi)||p(\zeta|z = 0)\right]$$

into $q(z|x, \phi)$. This is especially simple if $q(\zeta_i|z_i, x, \phi) = p(\zeta_i|z_i)$ when $z_i = 0$, since then $\mathrm{KL}\left[q(\zeta|z = 0, x, \phi)||p(\zeta|z = 0)\right] = 0$.

### E.1 Spike-and-Gaussian

We might wish $q(\zeta_i|z_i, x, \phi)$ to be a separate Gaussian for both values of the binary $z_i$. However, it is difficult to invert the CDF of the resulting mixture of Gaussians. It is much easier to use a mixture of a delta spike and a Gaussian, for which the CDF can inverted piecewise:

$$q(\zeta_i|z_i = 0, x, \phi) = \delta(\zeta_i) \qquad F_{q(\zeta_i|z_i = 0, x, \phi)}(\zeta_i) = H(\zeta_i) = \begin{cases} 0, & \text{if } \zeta_i < 0 \\ 1, & \text{otherwise} \end{cases}$$

$$q(\zeta_i|z_i = 1, x, \phi) = \mathcal{N}\left(\mu_{q,i}(x, \phi), \sigma_{q,i}^2(x, \phi)\right) \quad F_{q(\zeta_i|z_i = 1, x, \phi)}(\zeta_i) = \frac{1}{2}\left[1 + \mathrm{erf}\left(\frac{\zeta_i - \mu_{q,i}(x, \phi)}{\sqrt{2}\sigma_{q,i}(x, \phi)}\right)\right]$$

where $\mu_q(x, \phi)$ and $\sigma_q(x, \phi)$ are functions of $x$ and $\phi$. We use the substitutions $q(z_i = 1|x, \phi) \to q$, $\mu_{q,i}(x, \phi) \to \mu_{q,i}$, and $\sigma_{q,i}(x, \phi) \to \sigma_{q,i}$ in the sequel to simplify notation. The prior distribution $p$ is similarly parameterized.

We can now find the CDF for $q(\zeta|x, \phi)$ as a function of $q(z = 1|x, \phi) \to q$:

$$F_{q(\zeta|x,\phi)}(\zeta_i) = (1 - q_i) \cdot H(\zeta_i)$$

$$+ \frac{q_i}{2} \cdot \left[1 + \mathrm{erf}\left(\frac{\zeta_i - \mu_{q,i}}{\sqrt{2}\sigma_{q,i}}\right)\right]$$

Since $z_i = 0$ makes no contribution to the CDF until $\zeta_i = 0$, the value of $\rho$ at which $\zeta_i = 0$ is

$$\rho_i^{step} = \frac{q_i}{2}\left[1 + \mathrm{erf}\left(\frac{-\mu_{q,i}}{\sqrt{2}\sigma_{q,i}}\right)\right]$$

so:

$$\zeta_i = \begin{cases} \mu_{q,i} + \sqrt{2}\sigma_{q,i} \cdot \mathrm{erf}^{-1}\left(\frac{2\rho_i}{q_i} - 1\right), & \text{if } \rho_i < \rho_i^{step} \\ 0, & \text{if } \rho_i^{step} \leq \rho_i \leq \rho_i^{step} + (1 - q_i) \\ \mu_{q,i} + \sqrt{2}\sigma_{q,i} \cdot \mathrm{erf}^{-1}\left(\frac{2(\rho_i - 1)}{q_i} + 1\right), & \text{otherwise} \end{cases}$$

Gradients are always evaluated for fixed choices of $\rho$, and gradients are never taken with respect to $\rho$. As a result, expectations with respect to $\rho$ are invariant to permutations of $\rho$. Furthermore,

$$\frac{2\rho_i}{q_i} - 1 = \frac{2(\rho_i' - 1)}{q_i} + 1$$

where $\rho_i' = \rho_i + (1 - q_i)$. We can thus shift the delta spike to the beginning of the range of $\rho_i$, and use

$$\zeta_i = \begin{cases} 0, & \text{if } \rho_i \leq 1 - q_i \\ \mu_{q,i} + \sqrt{2}\sigma_{q,i} \cdot \mathrm{erf}^{-1}\left(\frac{2(\rho_i - 1)}{q_i} + 1\right), & \text{otherwise} \end{cases}$$

All parameters of the multivariate Gaussians should be trainable functions of $x$, and independent of $q$. The new term in Equation 22 is:

$$\sum_z q(z|x, \phi) \cdot \mathrm{KL}\left[q(\zeta|z, x, \phi)||p(\zeta|z)\right] =$$

$$\sum_{z,i} q(z_i = 1|x, \phi) \cdot \mathrm{KL}\left[q(\zeta_i|z_i = 1, x, \phi)||p(\zeta_i|z_i = 1)\right]$$

$$+ (1 - q(z_i = 1|x, \phi)) \cdot \mathrm{KL}\left[q(\zeta_i|z_i = 0, x, \phi)||p(\zeta_i|z_i = 0)\right]$$

If $z_i = 0$, then $q(\zeta_i|z_i = 0, x, \phi) = p(\zeta_i|z_i = 0, \theta)$, and $\mathrm{KL}\left[q(\zeta_i|z_i = 0, x, \phi)||p(\zeta_i|z_i = 0, \theta)\right] = 0$ as in Section 2. The KL divergence between two multivariate Gaussians with diagonal covariance matrices, with means $\mu_{p,i}$, $\mu_{q,i}$, and covariances $\sigma_{p,i}^2$ and $\sigma_{q,i}^2$, is

$$\mathrm{KL}\left[q||p\right] = \sum_i \left(\log \sigma_{p,i} - \log \sigma_{q,i} + \frac{\sigma_{q,i}^2 + (\mu_{q,i} - \mu_{p,i})^2}{2 \cdot \sigma_{p,i}^2} - \frac{1}{2}\right)$$

To train $q(z_i = 1|x, \phi)$, we thus need to backpropagate $\mathrm{KL}\left[q(\zeta_i|z_i = 1, x, \phi)||p(\zeta_i|z_i = 1)\right]$ into it.

Finally,

$$\frac{\partial \mathrm{KL}[q||p]}{\partial \mu_{q,i}} = \frac{\mu_{q,i} - \mu_{p,i}}{\sigma_{p,i}^2}$$

$$\frac{\partial \mathrm{KL}[q||p]}{\partial \sigma_{q,i}} = -\frac{1}{\sigma_{q,i}} + \frac{\sigma_{q,i}}{\sigma_{p,i}^2}$$

so

$$\sum_z q(z|x,\phi) \cdot \frac{\partial}{\partial \mu_{q,i}} \text{KL}\left[q||p\right] = q(z_i = 1|x,\phi) \cdot \frac{\mu_{q,i} - \mu_{p,i}}{\sigma_{p,i}^2}$$

$$\sum_z q(z|x,\phi) \cdot \frac{\partial}{\partial \sigma_{q,i}} \text{KL}\left[q||p\right] = q(z_i = 1|x,\phi) \cdot \left(-\frac{1}{\sigma_{q,i}} + \frac{\sigma_{q,i}}{\sigma_{p,i}^2}\right)$$

For $p$, it is not useful to make the mean values of $\zeta$ adjustable for each value of $z$, since this is redundant with the parameterization of the decoder. With fixed means, we could still parameterize the variance, but to maintain correspondence with the standard VAE, we choose the variance to be one.

## F   COMPUTING THE GRADIENT OF $\text{KL}\left[q(\zeta, z|x, \phi)||p(\zeta, z|\theta)\right]$

The KL term of the ELBO (Equation 2) is not significantly affected by the introduction of additional continuous latent variables $\zeta$, so long as we use the same expansion $r(\zeta|z)$ for both the approximating posterior and the prior:

$$\text{KL}\left[q||p\right] = \sum_z \int_\zeta \left(\prod_{1 \le j \le k} r(\zeta_j|z_j) \cdot q(z_j|\zeta_{i<j}, x)\right) \cdot \log\left[\frac{\prod_{1 \le j \le k} r(\zeta_j|z_j) \cdot q(z_j|\zeta_{i<j}, x)}{p(z) \cdot \prod_{1 \le j \le k} r(\zeta_j|z_j)}\right]$$

$$= \sum_z \int_\zeta \left(\prod_{1 \le j \le k} r(\zeta_j|z_j) \cdot q(z_j|\zeta_{i<j}, x)\right) \cdot \log\left[\frac{\prod_{1 \le j \le k} q(z_j|\zeta_{i<j}, x)}{p(z)}\right]. \quad (24)$$

The gradient of Equation 24 with respect to the parameters $\theta$ of the prior, $p(z|\theta)$, can be estimated stochastically using samples from the approximating posterior, $q(\zeta, z|x, \phi)$, and the true prior, $p(z|\theta)$. When the prior is an RBM, defined by Equation 6, we find:

$$-\frac{\partial}{\partial \theta} \text{KL}\left[q||p\right] = -\sum_{\zeta,z} q(\zeta, z|x, \phi) \cdot \frac{\partial E_p(z, \theta)}{\partial \theta} + \sum_z p(z|\theta) \cdot \frac{\partial E_p(z, \theta)}{\partial \theta}$$

$$= -\mathbb{E}_{q(z_1|x,\phi)}\left[\cdots\left[\mathbb{E}_{q(z_k|\zeta_{i<k}, x, \phi)}\left[\frac{\partial E_p(z, \theta)}{\partial \theta}\right]\right]\right] + \mathbb{E}_{p(z|\theta)}\left[\frac{\partial E_p(z, \theta)}{\partial \theta}\right] \quad (25)$$

The final expectation with respect to $q(z_k|\zeta_{i<k}, x, \phi)$ can be performed analytically; all other expectations require samples from the approximating posterior. Similarly, for the prior, we must sample from the RBM, although Rao-Blackwellization can be used to marginalize half of the units.

### F.1   GRADIENT OF THE ENTROPY WITH RESPECT TO $\phi$

In contrast, the gradient of the KL term with respect to the parameters of the approximating posterior is severely complicated by a nonfactorial approximating posterior. We break $\text{KL}\left[q||p\right]$ into two terms, the negative entropy $\sum_{z,\zeta} q \log q$, and the cross-entropy $-\sum_{z,\zeta} q \log p$, and compute their gradients separately.

We can regroup the negative entropy term of the KL divergence so as to use the reparameterization trick to backpropagate through $\prod_{i<j} q(z_j|\zeta_{i<j}, x)$:

$$-H(q) = \sum_z \int_\zeta \left( \prod_{1 \leq j \leq k} r(\zeta_j|z_j) \cdot q(z_j|\zeta_{i<j}, x) \right) \cdot \log \left[ \prod_{1 \leq j \leq k} q(z_j|\zeta_{i<j}, x) \right]$$

$$= \sum_z \int_\zeta \left( \prod_j r(\zeta_j|z_j) \cdot q(z_j|\zeta_{i<j}, x) \right) \cdot \left( \sum_j \log q(z_j|\zeta_{i<j}, x) \right)$$

$$= \sum_j \sum_z \int_\zeta \left( \prod_{i \leq j} r(\zeta_i|z_i) \cdot q(z_i|\zeta_{h<i}, x) \right) \cdot \log q(z_j|\zeta_{i<j}, x)$$

$$= \sum_j \mathbb{E}_{q(\zeta_{i<j}, z_{i<j}|x, \phi)} \left[ \sum_{z_j} q(z_j|\zeta_{i<j}, x) \cdot \log q(z_j|\zeta_{i<j}, x) \right].$$

$$= \sum_j \mathbb{E}_{\rho_{i<j}} \left[ \sum_{z_j} q(z_j|\rho_{i<j}, x) \cdot \log q(z_j|\rho_{i<j}, x) \right] \qquad (26)$$

where indices $i$ and $j$ denote hierarchical groups of variables. The probability $q(z_j|\rho_{i<j}, x)$ is evaluated analytically, whereas all variables $z_{i<j}$ and $\zeta_{i<j}$ are implicitly sampled stochastically via $\rho_{i<j}$.

We wish to take the gradient of $-H(q)$ in Equation 26. Using the identity:

$$\mathbb{E}_q \left[ c \cdot \frac{\partial}{\partial \phi} \log q \right] = c \cdot \sum_z q \cdot \left( \frac{\partial q}{\partial \phi} / q \right) = c \cdot \frac{\partial}{\partial \phi} \left( \sum_z q \right) = 0 \qquad (27)$$

for any constant $c$, we can eliminate the gradient of $\log q_{j|\rho_{i<j}}$ in $-\frac{\partial H(q)}{\partial \phi}$, and obtain:

$$-\frac{\partial}{\partial \phi} H(q) = \sum_j \mathbb{E}_{\rho_{i<j}} \left[ \sum_{z_j} \left( \frac{\partial}{\partial \phi} q(z_j|\rho_{i<j}, x) \right) \cdot \log q(z_j|\rho_{i<j}, x) \right].$$

Moreover, we can eliminate any log-partition function in $\log q(z_j|\rho_{i<j}, x)$ by an argument analogous to Equation 27.[15] By repeating this argument one more time, we can break $\frac{\partial}{\partial \phi} q(z_j|\rho_{i<j}, x)$ into its factorial components.[16] If $z_i \in \{0, 1\}$, then using Equation 10, gradient of the negative entropy reduces to:

$$-\frac{\partial}{\partial \phi} H(q) = \sum_j \mathbb{E}_{\rho_{i<j}} \left[ \sum_{\iota \in j} \sum_{z_\iota} q_\iota(z_\iota) \cdot \left( z_\iota \cdot \frac{\partial g_\iota}{\partial \phi} - \sum_{z_\iota} \left( q_\iota(z_\iota) \cdot z_\iota \cdot \frac{\partial g_\iota}{\partial \phi} \right) \right) \cdot (g_\iota \cdot z_\iota) \right]$$

$$= \sum_j \mathbb{E}_{\rho_{i<j}} \left[ \frac{\partial g_j^\top}{\partial \phi} \cdot \left( g_j \odot \left[ q_j(z_j = 1) - q_j^2(z_j = 1) \right] \right) \right]$$

where $\iota$ and $z_\iota$ correspond to single variables within the hierarchical groups denoted by $j$. In TensorFlow, it might be simpler to write:

$$-\frac{\partial}{\partial \phi} H(q) = \mathbb{E}_{\rho_{i<j}} \left[ \frac{\partial q_j^\top (z_j = 1)}{\partial \phi} \cdot g_j \right].$$

---

[15] $\sum_z c \cdot \frac{\partial q}{\partial \phi} = c \cdot \frac{\partial}{\partial \phi} \sum_z q = 0$, where $c$ is the log partition function of $q(z_j|\rho_{i<j}, x)$.

[16] $\frac{\partial}{\partial \phi} \prod_i q_i = \sum_i \frac{\partial q_i}{\partial \phi} \cdot \prod_{j \neq i} q_j$, so the $q_{j \neq i}$ marginalize out of $\frac{\partial q_i}{\partial \phi} \cdot \prod_{j \neq i} q_j$ when multiplied by $\log q_i$. When $\frac{\partial q_i}{\partial \phi} \cdot \prod_{j \neq i} q_j$ is multiplied by one of the $\log q_{j \neq i}$, the sum over $z_i$ can be taken inside the $\frac{\partial}{\partial \phi}$, and again $\frac{\partial}{\partial \phi} \sum_{z_i} q_i = 0$.

## F.2 GRADIENT OF THE CROSS-ENTROPY

The gradient of the cross-entropy with respect to the parameters $\phi$ of the approximating posterior does not depend on the partition function of the prior $\mathcal{Z}_p$, since:

$$-\frac{\partial}{\partial \phi} \sum_z q \log p = \sum_z \frac{\partial}{\partial \phi} q \cdot E_p + \frac{\partial}{\partial \phi} q \cdot \log \mathcal{Z}_p = \sum_z \frac{\partial}{\partial \phi} q \cdot E_p$$

by Equations 6 and 27, so we are left with the gradient of the average energy $E_p$.

The remaining cross-entropy term is

$$\sum_z q \cdot E_p = -\mathbb{E}_\rho \left[ z^\top \cdot W \cdot z + b^\top \cdot z \right].$$

We can handle the term $b^\top \cdot z$ analytically, since $z_i \in \{0, 1\}$, and

$$\mathbb{E}_\rho \left[ b^\top \cdot z \right] = b^\top \cdot \mathbb{E}_\rho \left[ q(z = 1) \right].$$

The approximating posterior $q$ is continuous, with nonzero derivative, so the reparameterization trick can be applied to backpropagate gradients:

$$\frac{\partial}{\partial \phi} \mathbb{E}_\rho \left[ b^\top \cdot z \right] = b^\top \cdot \mathbb{E}_\rho \left[ \frac{\partial}{\partial \phi} q(z = 1) \right].$$

In contrast, each element of the sum

$$z^\top \cdot W \cdot z = \sum_{i,j} W_{ij} \cdot z_i \cdot z_j$$

depends upon variables that are not usually in the same hierarchical level, so in general

$$\mathbb{E}_\rho \left[ W_{ij} z_i z_j \right] \neq W_{ij} \mathbb{E}_\rho \left[ z_i \right] \cdot \mathbb{E}_\rho \left[ z_j \right].$$

We might decompose this term into

$$\mathbb{E}_\rho \left[ W_{ij} z_i z_j \right] = W_{ij} \cdot \mathbb{E}_{\rho_{k \leq i}} \left[ z_i \cdot \mathbb{E}_{\rho_{k > i}} \left[ z_j \right] \right],$$

where without loss of generality $z_i$ is in an earlier hierarchical layer than $z_j$; however, it is not clear how to take the derivative of $z_i$, since it is a discontinuous function of $\rho_{k \leq i}$.

## F.3 NAIVE APPROACH

The naive approach would be to take the gradient of the expectation using the gradient of log-probabilities over all variables:

$$\frac{\partial}{\partial \phi} \mathbb{E} \left[ W_{ij} z_i z_j \right] = \mathbb{E}_q \left[ W_{ij} z_i z_j \cdot \frac{\partial}{\partial \phi} \log q \right]$$

$$= \mathbb{E}_{q_1, q_{2|1}, \dots} \left[ W_{ij} z_i z_j \cdot \sum_k \frac{\partial}{\partial \phi} \log q_{k|l<k} \right] \qquad (28)$$

$$= \mathbb{E}_{q_1, q_{2|1}, \dots} \left[ W_{ij} z_i z_j \cdot \sum_k \frac{1}{q_{k|l<k}} \cdot \frac{\partial q_{k|l<k}}{\partial \phi} \right].$$

For $\frac{\partial q_{k|l<k}}{\partial \phi}$, we can drop out terms involving only $z_{i<k}$ and $z_{j<k}$ that occur hierarchically before $k$, since those terms can be pulled out of the expectation over $q_k$, and we can apply Equation 27. However, for terms involving $z_{i>k}$ or $z_{j>k}$ that occur hierarchically after $k$, the expected value of $z_i$ or $z_j$ depends upon the chosen value of $z_k$.

The gradient calculation in Equation 28 is an instance of the REINFORCE algorithm (Equation 18). Moreover, the variance of the estimate is proportional to the number of terms (to the extent that the terms are independent). The number of terms contributing to each gradient $\frac{\partial q_{k|l<k}}{\partial \phi}$ grows quadratically with number of units in the RBM. We can introduce a baseline, as in NVIL (Mnih & Gregor, 2014):

$$\mathbb{E}_q \left[ (W_{ij} z_i z_j - c(x)) \cdot \frac{\partial}{\partial \phi} \log q \right],$$

but this approximation is still high-variance.

## F.4 DECOMPOSITION OF $\frac{\partial}{\partial \phi} W_{ij} z_i z_j$ VIA THE CHAIN RULE

When using the spike-and-exponential, spike-and-slab, or spike-and-Gaussian distributions of sections 2.1 D.2, and E.1, we can decompose the gradient of $\mathbb{E}\left[W_{ij} z_i z_j\right]$ using the chain rule. Previously, we have considered $z$ to be a function of $\rho$ and $\phi$. We can instead formulate $z$ as a function of $q(z=1)$ and $\rho$, where $q(z=1)$ is itself a function of $\rho$ and $\phi$. Specifically,

$$z_i(q_i(z_i=1), \rho_i) = \begin{cases} 0 & \text{if } \rho_i < 1 - q_i(z_i=1) = q_i(z_i=0) \\ 1 & \text{otherwise.} \end{cases} \tag{29}$$

Using the chain rule, $\frac{\partial}{\partial \phi} z_i = \sum_j \frac{\partial z_i}{\partial q_j(z_j=1)} \cdot \frac{\partial q_j(z_j=1)}{\partial \phi}$, where $\frac{\partial z_i}{\partial q_j(z_j=1)}$ holds all $q_{k \neq j}$ fixed, even though they all depend on the common variables $\rho$ and parameters $\phi$. We use the chain rule to differentiate with respect to $q(z=1)$ since it allows us to pull part of the integral over $\rho$ inside the derivative with respect to $\phi$. In the sequel, we sometimes write $q$ in place of $q(z=1)$ to minimize notational clutter.

Expanding the desired gradient using the reparameterization trick and the chain rule, we find:

$$\frac{\partial}{\partial \phi} \mathbb{E}_q\left[W_{ij} z_i z_j\right] = \frac{\partial}{\partial \phi} \mathbb{E}_\rho\left[W_{ij} z_i z_j\right]$$
$$= \mathbb{E}_\rho\left[\sum_k \frac{\partial W_{ij} z_i z_j}{\partial q_k(z_k=1)} \cdot \frac{\partial q_k(z_k=1)}{\partial \phi}\right]. \tag{30}$$

We can change the order of integration (via the expectation) and differentiation since

$$|W_{ij} z_i z_j| \leq W_{ij} < \infty$$

for all $\rho$ and bounded $\phi$ (Cheng, 2006). Although $z(q, \rho)$ is a step function, and its derivative is a delta function, the integral (corresponding to the expectation with respect to $\rho$) of its derivative is finite. Rather than dealing with generalized functions directly, we apply the definition of the derivative, and push through the matching integral to recover a finite quantity.

For simplicity, we pull the sum over $k$ out of the expectation in Equation 30, and consider each summand independently. From Equation 29, we see that $z_i$ is only a function of $q_i$, so all terms in the sum over $k$ in Equation 30 vanish except $k = i$ and $k = j$. Without loss of generality, we consider the term $k = i$; the term $k = j$ is symmetric. Applying the definition of the gradient to one of the summands, and then analytically taking the expectation with respect to $\rho_i$, we obtain:

$$\mathbb{E}_\rho\left[\frac{\partial W_{ij} \cdot z_i(q, \rho) \cdot z_j(q, \rho)}{\partial q_i(z_i=1)} \cdot \frac{\partial q_i(z_i=1)}{\partial \phi}\right]$$
$$= \mathbb{E}_\rho\left[\lim_{\delta q_i(z_i=1) \to 0} \frac{W_{ij} \cdot z_i(q+\delta q_i, \rho) \cdot z_j(q+\delta q_i, \rho) - W_{ij} \cdot z_i(q, \rho) \cdot z_j(q, \rho)}{\delta q_i} \cdot \frac{\partial q_i(z_i=1)}{\partial \phi}\right]$$
$$= \mathbb{E}_{\rho_{k \neq i}}\left[\lim_{\delta q_i(z_i=1) \to 0} \delta q_i \cdot \frac{W_{ij} \cdot 1 \cdot z_j(q, \rho) - W_{ij} \cdot 0 \cdot z_j(q, \rho)}{\delta q_i} \cdot \frac{\partial q_i(z_i=1)}{\partial \phi}\bigg|_{\rho_i = q_i(z_i=0)}\right]$$
$$= \mathbb{E}_{\rho_{k \neq i}}\left[W_{ij} \cdot z_j(q, \rho) \cdot \frac{\partial q_i(z_i=1)}{\partial \phi}\bigg|_{\rho_i = q_i(z_i=0)}\right].$$

The third line follows from Equation 29, since $z_i(q+\delta q_i, \rho)$ differs from $z_i(q, \rho)$ only in the region of $\rho$ of size $\delta q_i$ around $q_i(z_i=0) = 1 - q_i(z_i=1)$ where $z_i(q+\delta q_i, \rho) \neq z_i(q, \rho)$. Regardless of the choice of $\rho$, $z_j(q+\delta q_i, \rho) = z_j(q, \rho)$.

The third line fixes $\rho_i$ to the transition between $z_i = 0$ and $z_i = 1$ at $q_i(z_i=0)$. Since $z_i = 0$ implies $\zeta_i = 0$,[17] and $\zeta$ is a continuous function of $\rho$, the third line implies that $\zeta_i = 0$. At the same time, since $q_i$ is only a function of $\rho_{k<i}$ from earlier in the hierarchy, the term $\frac{\partial q_i}{\partial \phi}$ is not affected by the choice of $\rho_i$.[18] As noted above, due to the chain rule, the perturbation $\delta q_i$ has no effect on other

---

[17] We chose the conditional distribution $r(\zeta_i|z_i=0)$ to be a delta spike at zero.

[18] In contrast, $z_i$ *is* a function of $\rho_i$.

$q_j$ by definition; the gradient is evaluated with those values held constant. On the other hand, $\frac{\partial q_i}{\partial \phi}$ is generally nonzero for all parameters governing hierarchical levels $k < i$.

Since $\rho_i$ is fixed such that $\zeta_i = 0$, all units further down the hierarchy must be sampled consistent with this restriction. A sample from $\rho$ has $\zeta_i = 0$ if $z_i = 0$, which occurs with probability $q_i(z_i = 0)$.[19] We can compute the gradient with a stochastic approximation by multiplying each sample by $1 - z_i$, so that terms with $\zeta_i \neq 0$ are ignored,[20] and scaling up the gradient when $z_i = 0$ by $\frac{1}{q_i(z_i=0)}$:

$$\frac{\partial}{\partial \phi} \mathbb{E}\left[W_{ij} z_i z_j\right] = \mathbb{E}_\rho \left[ W_{ij} \cdot \frac{1 - z_i}{1 - q_i(z_i = 1)} \cdot z_j \cdot \frac{\partial q_i(z_i = 1)}{\partial \phi} \right]. \tag{31}$$

The term $\frac{1-z_i}{1-q_i}$ is not necessary if $j$ comes before $i$ in the hierarchy.

While Equation 31 appears similar to REINFORCE, it is better understood as an importance-weighted estimate of an efficient gradient calculation. Just as a ReLU only has a nonzero gradient in the linear regime, $\frac{\partial z_i}{\partial \phi}$ effectively only has a nonzero gradient when $z_i = 0$, in which case $\frac{\partial z_i}{\partial \phi} \sim \frac{\partial q_i(z_i=1)}{\partial \phi}$. Unlike in REINFORCE, we do effectively differentiate the reward, $W_{ij} z_i z_j$. Moreover, the number of terms contributing to each gradient $\frac{\partial q_i(z_i=1)}{\partial \phi}$ grows linearly with the number of units in an RBM, whereas it grows quadratically in the method of Section F.3.

# G   MOTIVATION FOR BUILDING APPROXIMATING POSTERIOR AND PRIOR HIERARCHIES IN THE SAME ORDER

Intuition regarding the difficulty of approximating the posterior distribution over the latent variables given the data can be developed by considering sparse coding, an approach that uses a basis set of spatially locallized filters (Olshausen & Field, 1996). The basis set is overcomplete, and there are generally many basis elements similar to any selected basis element. However, the sparsity prior pushes the posterior distribution to use only one amongst each set of similar basis elements.

As a result, there is a large set of sparse representations of roughly equivalent quality for any single input. Each basis element individually can be replaced with a similar basis element. However, having changed one basis element, the optimal choice for the adjacent elements also changes so the filters mesh properly, avoiding redundancy or gaps. The true posterior is thus highly correlated, since even after conditioning on the input, the probability of a given basis element depends strongly on the selection of the adjacent basis elements.

These equivalent representations can easily be disambiguated by the successive layers of the representation. In the simplest case, the previous layer could directly specify which correlated set of basis elements to use amongst the applicable sets. We can therefore achieve greater efficiency by inferring the approximating posterior over the top-most latent layer first. Only then do we compute the conditional approximating posteriors of lower layers given a sample from the approximating posterior of the higher layers, breaking the symmetry between representations of similar quality.

# H   ARCHITECTURE

The stochastic approximation to the ELBO is computed via one pass down the approximating posterior (Figure 4a), sampling from each continuous latent layer $\zeta_i$ and $\mathfrak{z}_{m>1}$ in turn; and another pass down the prior (Figure 4b), conditioned on the sample from the approximating posterior. In the pass down the prior, signals do not flow from layer to layer through the entire model. Rather, the input to each layer is determined by the approximating posterior of the previous layers, as follows from Equation 14. The gradient is computed by backpropagating the reconstruction log-likelihood, and the KL divergence between the approximating posterior and true prior at each layer, through this differentiable structure.

---

[19]It might also be the case that $\zeta_i = 0$ when $z_i = 1$, but with our choice of $r(\zeta|z)$, this has vanishingly small probability.

[20]This takes advantage of the fact that $z_i \in \{0, 1\}$.

All hyperparameters were tuned via manual experimentation. Except in Figure 6, RBMs have 128 units (64 units per side, with full bipartite connections between the two sides), with 4 layers of hierarchy in the approximating posterior. We use 100 iterations of block Gibbs sampling, with 20 persistent chains per element of the minibatch, to sample from the prior in the stochastic approximation to Equation 11.

When using the hierarchy of continuous latent variables described in Section 4, discrete VAEs overfit if any component of the prior is overparameterized, as shown in Figure 9a. In contrast, a larger and more powerful approximating posterior generally did not reduce performance within the range examined, as in Figure 9b. In response, we manually tuned the number of layers of continuous latent variables, the number of such continuous latent variables per layer, the number of deterministic hidden units per layer in the neural network defining each hierarchical layer of the prior, and the use of parameter sharing in the prior. We list the selected values in Table 2. All neural networks implementing components of the approximating posterior contain two hidden layers of 2000 units.

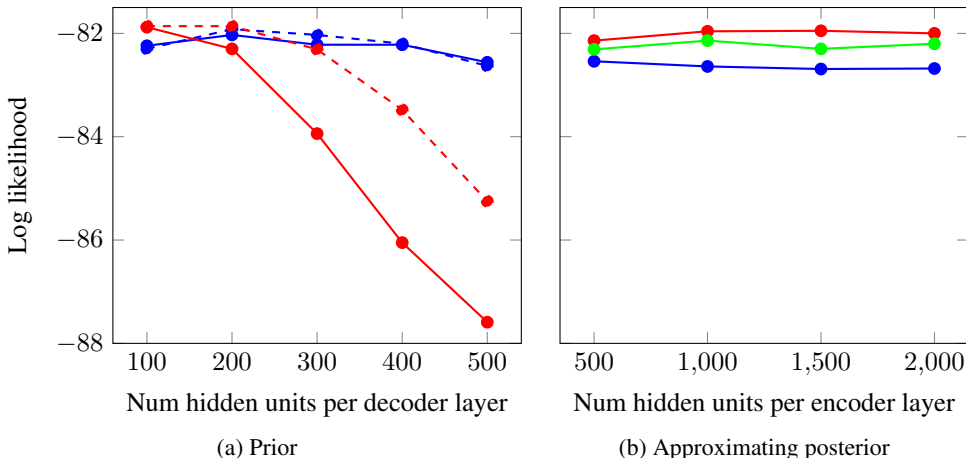

(a) Prior  (b) Approximating posterior

Figure 9: Log likelihood on statically binarized MNIST versus the number of hidden units per neural network layer, in the prior (a) and approximating posterior (b). The number of deterministic hidden layers in the networks parameterizing the prior/approximating posterior is 1 (blue), 2 (red), 3 (green) in (a/b), respectively. The number of deterministic hidden layers in the final network parameterizing $p(x|\mathfrak{z})$ is 0 (solid) or 1 (dashed). All models use only 10 layers of continuous latent variables, with no parameter sharing.

| | Num layers | Vars per layer | Hids per prior layer | Param sharing |
|---|---|---|---|---|
| MNIST (dyn bin) | 18 | 64 | 1000 | none |
| MNIST (static bin) | 20 | 256 | 2000 | 2 groups |
| Omniglot | 16 | 256 | 800 | 2 groups |
| Caltech-101 Sil | 12 | 80 | 100 | complete |

Table 2: Architectural hyperparameters used for each dataset. Successive columns list the number of layers of continuous latent variables, the number of such continuous latent variables per layer, the number of deterministic hidden units per layer in the neural network defining each hierarchical layer of the prior, and the use of parameter sharing in the prior. Smaller datasets require more regularization, and achieve optimal performance with a smaller prior.

On statically binarized MNIST, Omniglot, and Caltech-101 Silhouettes, we further regularize using recurrent parameter sharing. In the simplest case, each $p(\mathfrak{z}_m|\mathfrak{z}_{l<m}, \theta)$ and $p(x|\mathfrak{z}, \theta)$ is a function of $\sum_{l<m} \mathfrak{z}_l$, rather than a function of the concatenation $[\mathfrak{z}_0, \mathfrak{z}_1, \ldots, \mathfrak{z}_{m-1}]$. Moreover, all $p(\mathfrak{z}_{m\geq 1}|\mathfrak{z}_{l<m}, \theta)$ share parameters. The RBM layer $\mathfrak{z}_0$ is rendered compatible with this parameterization by using a trainable linear transformation of $\zeta$, $M \cdot \zeta$; where the number of rows in $M$ is

equal to the number of variables in each $\mathfrak{z}_{m>0}$. We refer to this architecture as complete recurrent parameter sharing.

On datasets of intermediate size, a degree of recurrent parameter sharing somewhere between full independence and complete sharing is beneficial. We define the $n$ group architecture by dividing the continuous latent layers $\mathfrak{z}_{m \geq 1}$ into $n$ equally sized groups of consecutive layers. Each such group is independently subject to recurrent parameter sharing analogous to the complete sharing architecture, and the RBM layer $\mathfrak{z}_0$ is independently parameterized.

We use the spike-and-exponential transformation described in Section 2.1. The exponent is a trainable parameter, but it is bounded above by a value that increases linearly with the number of training epochs. We use warm-up with strength 20 for 5 epochs, and additional warm-up of strength 2 on the RBM alone for 20 epochs (Raiko et al., 2007; Bowman et al., 2016; Sønderby et al., 2016).

When $p(x|\mathfrak{z})$ is linear, all nonlinear transformations are part of the prior over the latent variables. In contrast, it is also possible to define the prior distribution over the continuous latent variables to be a simple factorial distribution, and push the nonlinearity into the final decoder $p(x|\mathfrak{z})$, as in traditional VAEs. The former case can be reduced to something analogous to the latter case using the reparameterization trick.

However, a VAE with a completely independent prior does not regularize the nonlinearity of the prior; whereas a hierarchical prior requires that the nonlinearity of the prior (via its effect on the true posterior) be well-represented by the approximating posterior. Viewed another way, a completely independent prior requires the model to consist of many independent sources of variance, so the data manifold must be fully unfolded into an isotropic ball. A hierarchical prior allows the data manifold to remain curled within a higher-dimensional ambient space, with the approximating posterior merely tracking its contortions. A higher-dimensional ambient space makes sense when modeling multiple classes of objects. For instance, the parameters characterizing limb positions and orientations for people have no analog for houses.

## H.1 Estimating the log partition function

We estimate the log-likelihood by subtracting an estimate of the log partition function of the RBM ($\log \mathcal{Z}_p$ from Equation 6) from an importance-weighted computation analogous to that of Burda et al. (2016). For this purpose, we estimate the log partition function using bridge sampling, a variant of Bennett's acceptance ratio method (Bennett, 1976; Shirts & Chodera, 2008), which produces unbiased estimates of the partition function. Interpolating distributions were of the form $p(x)^\beta$, and sampled with a parallel tempering routine (Swendsen & Wang, 1986). The set of smoothing parameters $\beta$ in $[0, 1]$ were chosen to approximately equalize replica exchange rates at 0.5. This standard criteria simultaneously keeps mixing times small, and allows for robust inference. We make a conservative estimate for burn-in (0.5 of total run time), and choose the total length of run, and number of repeated experiments, to achieve sufficient statistical accuracy in the log partition function. In Figure 10, we plot the distribution of independent estimations of the log-partition function for a single model of each dataset. These estimates differ by no more than about 0.1, indicating that the estimate of the log-likelihood should be accurate to within about 0.05 nats.

## H.2 Constrained Laplacian batch normalization

Rather than traditional batch normalization (Ioffe & Szegedy, 2015), we base our batch normalization on the L1 norm. Specifically, we use:

$$\mathbf{y} = \mathbf{x} - \overline{\mathbf{x}}$$
$$\mathbf{x}_{bn} = \mathbf{y} / \left( \overline{|\mathbf{y}|} + \epsilon \right) \odot \mathbf{s} + \mathbf{o},$$

where $\mathbf{x}$ is a minibatch of scalar values, $\overline{\mathbf{x}}$ denotes the mean of $\mathbf{x}$, $\odot$ indicates element-wise multiplication, $\epsilon$ is a small positive constant, $\mathbf{s}$ is a learned scale, and $\mathbf{o}$ is a learned offset. For the approximating posterior over the RBM units, we bound $2 \leq \mathbf{s} \leq 3$, and $-\mathbf{s} \leq \mathbf{o} \leq \mathbf{s}$. This helps ensure that all units are both active and inactive in each minibatch, and thus that all units are used.

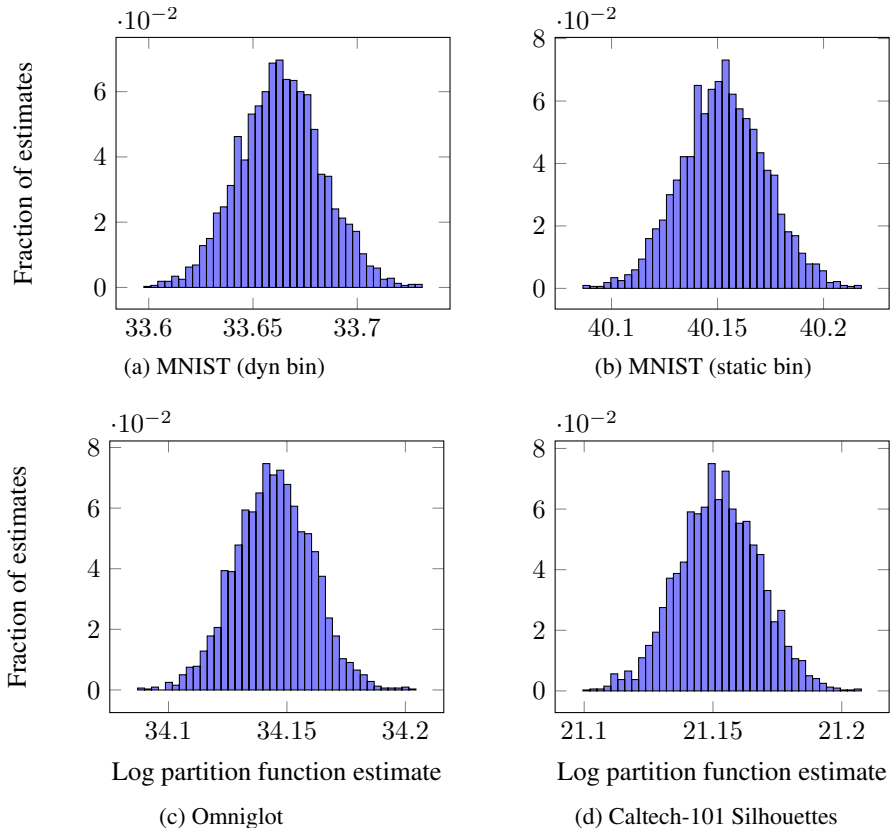

Figure 10: Distribution of estimates of the log-partition function, using Bennett's acceptance ratio method with parallel tempering, for a single model trained on dynamically binarized MNIST (a), statically binarized MNIST (b), Omniglot (c), and Caltech-101 Silhouettes (d)

## I    COMPARISON MODELS

In Table 1, we compare the performance of the discrete variational autoencoder to a selection of recent, competitive models. For dynamically binarized MNIST, we compare to deep belief networks (DBN; Hinton et al., 2006), reporting the results of Murray & Salakhutdinov (2009); importance-weighted autoencoders (IWAE; Burda et al., 2016); and ladder variational autoencoders (Ladder VAE; Sønderby et al., 2016).

For the static MNIST binarization of (Salakhutdinov & Murray, 2008), we compare to Hamiltonian variational inference (HVI; Salimans et al., 2015); the deep recurrent attentive writer (DRAW; Gregor et al., 2015); the neural adaptive importance sampler with neural autoregressive distribution estimator (NAIS NADE; Du et al., 2015); deep latent Gaussian models with normalizing flows (Normalizing flows; Rezende & Mohamed, 2015); and the variational Gaussian process (Tran et al., 2016).

On Omniglot, we compare to the importance-weighted autoencoder (IWAE; Burda et al., 2016); ladder variational autoencoder (Ladder VAE; Sønderby et al., 2016); and the restricted Boltzmann machine (RBM; Smolensky, 1986) and deep belief network (DBN; Hinton et al., 2006), reporting the results of Burda et al. (2015).

Finally, for Caltech-101 Silhouettes, we compare to the importance-weighted autoencoder (IWAE; Burda et al., 2016), reporting the results of Li & Turner (2016); reweighted wake-sleep with a deep sigmoid belief network (RWS SBN; Bornschein & Bengio, 2015); the restricted Boltzmann machine (RBM; Smolensky, 1986), reporting the results of Cho et al. (2013); and the neural adaptive importance sampler with neural autoregressive distribution estimator (NAIS NADE; Du et al., 2015).

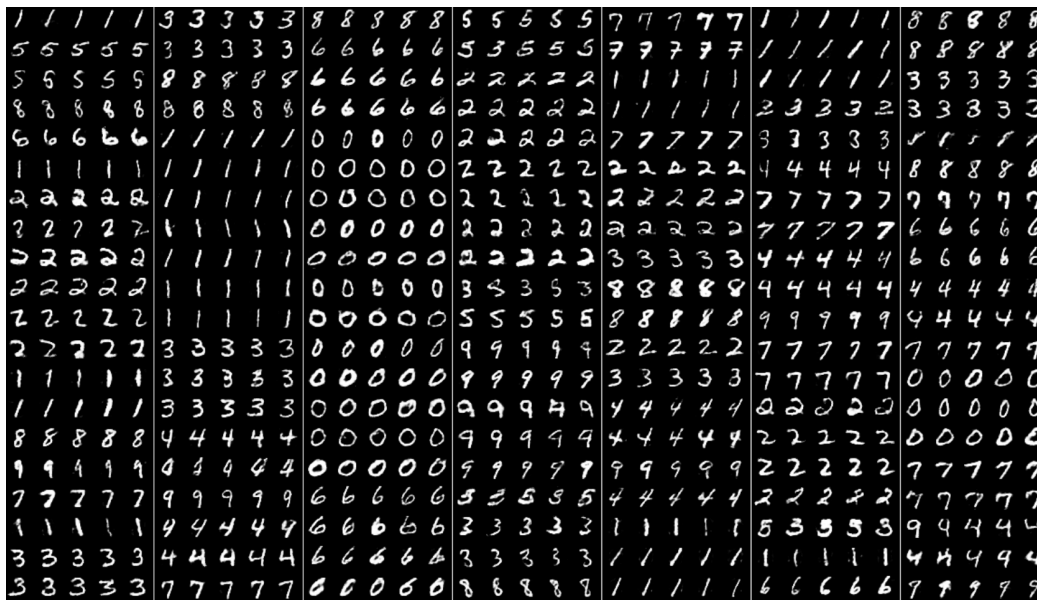

Figure 11: Evolution of samples from a discrete VAE trained on statically binarized MNIST, using persistent RBM Markov chains. We perform 100 iterations of block-Gibbs sampling on the RBM between successive rows. Each horizontal group of 5 uses a single, shared sample from the RBM, but independent continuous latent variables, and shows the variation induced by the continuous layers as opposed to the RBM. Vertical sequences in which the digit ID remains constant demonstrate that the RBM has distinct modes, each of which corresponds to a single digit ID, despite being trained in a wholly unsupervised manner.

## J SUPPLEMENTARY RESULTS

To highlight the contribution of the various components of our generative model, we investigate performance on a selection of simplified models.[21] First, we remove the continuous latent layers. The resulting prior, depicted in Figure 1b, consists of the bipartite Boltzmann machine (RBM), the smoothing variables $\zeta$, and a factorial Bernoulli distribution over the observed variables $x$ defined via a deep neural network with a logistic final layer. This probabilistic model achieves a log-likelihood of $-86.9$ with 128 RBM units and $-85.2$ with 200 RBM units.

Next, we further restrict the neural network defining the distribution over the observed variables $x$ given the smoothing variables $\zeta$ to consist of a linear transformation followed by a pointwise logistic nonlinearity, analogous to a sigmoid belief network (SBN; Spiegelhalter & Lauritzen, 1990; Neal, 1992). This decreases the negative log-likelihood to $-92.7$ with 128 RBM units and $-88.8$ with 200 RBM units.

We then remove the lateral connections in the RBM, reducing it to a set of independent binary random variables. The resulting network is a noisy sigmoid belief network. That is, samples are produced by drawing samples from the independent binary random variables, multiplying by an independent noise source, and then sampling from the observed variables as in a standard SBN. With this SBN-like architecture, the discrete variational autoencoder achieves a log-likelihood of $-97.0$ with 200 binary latent variables.

Finally, we replace the hierarchical approximating posterior of Figure 3a with the factorial approximating posterior of Figure 1a. This simplification of the approximating posterior, in addition to the prior, reduces the log-likelihood to $-102.9$ with 200 binary latent variables.

---

[21]In all cases, we report the negative log-likelihood on statically binarized MNIST (Salakhutdinov & Murray, 2008), estimated with $10^4$ importance weighted samples (Burda et al., 2016).

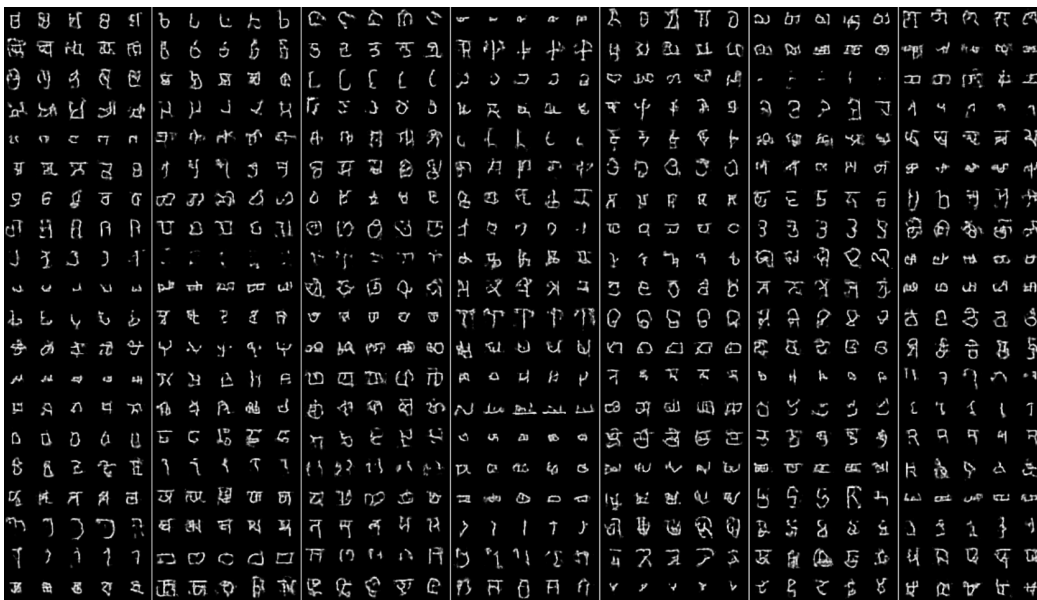

Figure 12: Evolution of samples from a discrete VAE trained on Omniglot, using persistent RBM Markov chains. We perform 100 iterations of block-Gibbs sampling on the RBM between successive rows. Each horizontal group of 5 uses a single, shared sample from the RBM, but independent continuous latent variables, and shows the variation induced by the continuous layers as opposed to the RBM.

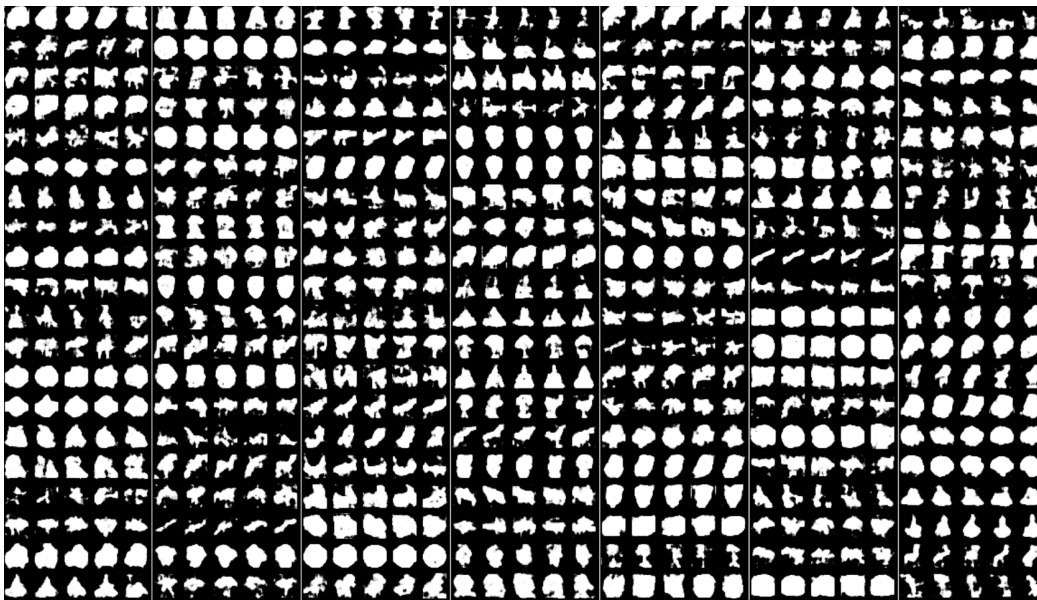

Figure 13: Evolution of samples from a discrete VAE trained on Caltech-101 Silhouettes, using persistent RBM Markov chains. We perform 100 iterations of block-Gibbs sampling on the RBM between successive rows. Each horizontal group of 5 uses a single, shared sample from the RBM, but independent continuous latent variables, and shows the variation induced by the continuous layers as opposed to the RBM. Vertical sequences in which the silhouette shape remains similar demonstrate that the RBM has distinct modes, each of which corresponds to a single silhouette type, despite being trained in a wholly unsupervised manner.

Figures 11, 12, and 13 repeat the analysis of Figure 5 for statically binarized MNIST, Omniglot, and Caltech-101 Silhouettes. Specifically, they show the generative output of a discrete VAE as the Markov chain over the RBM evolves via block Gibbs sampling. The RBM is held constant across each sub-row of five samples, and variation amongst these samples is due to the layers of continuous latent variables. Given a multimodal distribution with well-separated modes, Gibbs sampling passes through the large, low-probability space between the modes only infrequently. As a result, consistency of the object class over many successive rows in Figures 11, 12, and 13 indicates that the RBM prior has well-separated modes.

On statically binarized MNIST, the RBM still learns distinct, separated modes corresponding to most of the different digit types. However, these modes are not as well separated as in dynamically binarized MNIST, as is evident from the more rapid switching between digit types in Figure 11. There are not obvious modes for Omniglot in Figure 12; it is plausible that an RBM with 128 units could not represent enough well-separated modes to capture the large number of distinct character types in the Omniglot dataset. On Caltech-101 Silhouettes, there may be a mode corresponding to large, roughly convex blobs.

