# Peer review of "Discrete Variational Autoencoders"

_ICLR 2017 — accepted_

[Public Comment · (anonymous) · 28 Nov 2016]
**Gradient of the likelihood term**

This looked like an exciting though mathematically dense paper, so I spent a couple of days working through it. The presentation in the paper is not very clear and is in places incomplete, so I hope the author will correct me if I misunderstood some of the details of the proposed method.

As far as I can tell, the gradients of the variational objective w.r.t. the parameters phi of the variational posterior q(z, zeta) are incorrect, due to some of the probabilistic dependences being ignored. Most significantly, the gradient w.r.t parameters of q(z|x) seems to contain only the term obtained by differentiating -KL(q||p), but no contribution from the term E_q[log P(x|zeta)] in the objective. This likelihood term is considerably harder to deal with than the KL term as it does not factorize, so it is important to explain how it is taken into account when updating parameters of q(z|x). Ignoring this term amounts to training q(z|x) to match the prior p(z), teaching it to ignore the current observation x (assuming a factorial q(zeta|z) and r(zeta|z) that does not depend on x). This makes q(z|x) a completely ineffective recognition model.

Note that making r dependent on x or using a hierarchical posterior does allow q(z|x) to learn some dependence on x as z_i will depend on zeta_{j

[Public Comment · (anonymous) · 30 Nov 2016]
**Results without additional latent layers**

Have you tried training models with the smoothed RBM prior but without the Gaussian latent layers between it and the data? This seems like a crucial experiment to perform if you want to highlight the effectiveness of the aspects of your model that make it different from a VAE.

[Official Review · AnonReviewer2 · rating 8 · confidence 4 · 14 Dec 2016]
**Rich set of ideas on how to make VAEs work better.**

This is an interesting paper on how to handle reparameterization in VAEs when you have discrete variables. The idea is to introduce a smoothing transformation that is shared between the generative model and the recognition model (leading to cancellations). 
A second contribution is to introduce an RBM as the prior model P(z) and to use autoregressive connections in generative and recognition models. The whole package becomes a bit entangled and complex and it is hard to figure out what causes the claimed good performance. Experiments that study these contributions separately would have been nice. 
The framework does become a little complex but this should not be a problem if nice software is delivered that can be used in a plug and play mode.
Overall, the paper is very rich with ideas so I think it would be a great contribution to the conference.

[Public Comment · (anonymous) · 15 Dec 2016]
**Missing baselines**

This paper proposes a complex new architecture with some discrete latent variables along with a specialized method for training them. The trouble is that it's unclear how effective the method actually is at training the discrete part of the system. Is it better or worse at this than the existing methods?There're many such methods to choose from:
Hinton, G. E., Dayan, P., Frey, B. J., & Neal, R. M. (1995). The "wake-sleep" algorithm for unsupervised neural networks. 
Bengio, Y., Leonard, N., & Courville, A. (2013). Estimating or propagating gradients through stochastic neurons for conditional computation.
Gregor, K., Danihelka, I., Mnih, A., Blundell, C., & Wierstra, D. (2014). Deep autoregressive networks.
Mnih, A., & Gregor, K. (2014). Neural variational inference and learning in belief networks. 
Bornschein, J., & Bengio, Y. (2014). Reweighted wake-sleep.
Gu, S., Levine, S., Sutskever, I., & Mnih, A. (2015). MuProp: Unbiased Backpropagation for Stochastic Neural Networks.
Mnih, A., & Rezende, D. J. (2016). Variational inference for Monte Carlo objectives.
All of these algorithms have been used to train models with hundreds of variables, so they should be able to handle models with up to 128 discrete variables used in the paper easily, as long as the reparameterization trick is used for the Gaussian latent variables. Moreover, the above methods are applicable to any architecture with discrete latent variables, unlike the algorithm proposed in the paper, which can only be used if the discrete variables are smoothed out with the continuous ones. Why not compare to at least one of these more general algorithms? It would be good to know whether anything is gained by taking advantage of the model structure in this case.

[Official Review · AnonReviewer1 · rating 8 · confidence 2 · 19 Dec 2016]

Paper proposes a novel Variational Encoder architecture that contains discrete variables. Model contains an undirected discrete component that captures distribution over disconnected manifolds and a directed hierarchical continuous component that models the actual manifolds (induced by the discrete variables). In essence the model clusters the data and at the same time learns a continuous manifold representation for the clusters. The training procedure for such models is also presented and is quite involved. Experiments illustrate state-of-the-art performance on public datasets (including MNIST, Omniglot, Caltech-101). 

Overall the model is interesting and could be useful in a variety of applications and domains. The approach is complex and somewhat mathematically involved. It's not exactly clear how the model compares or relates to other RBM formulations, particularly those that contain discrete latent variables and continuous outputs. As a prime example:

Graham Taylor and Geoffrey Hinton. Factored conditional restricted Boltzmann machines for modeling motion style. In Proc. of the 26th International Conference on Machine Learning (ICML), 1025–1032, 2009.

Discussion of this should certainly be added.

[Official Review · AnonReviewer3 · rating 9 · confidence 4 · 19 Dec 2016]
**clever and useful contribution; clear and thorough exposition**

This paper presents a way of training deep generative models with discrete hidden variables using the reparameterization trick. It then applies it to a particular DBN-like architecture, and shows that this architecture achieves state-of-the-art density modeling performance on MNIST and similar datasets. 

The paper is well written, and the exposition is both thorough and precise. There are several appendices which justify various design decisions in detail. I wish more papers in our field would take this degree of care with the exposition!

The log-likelihood results are quite strong, especially given that most of the competitive algorithms are based on continuous latent variables. Probably the main thing missing from the experiments is some way to separate out the contributions of the architecture and the inference algorithm. (E.g., what if a comparable architecture is trained with VIMCO, or if the algorithm is applied to a previously published discrete architecture?)

I’m a bit concerned about the variance of the gradients in the general formulation of the algorithm. See my comment “variance of the derivatives of F^{-1}” below. I think the response is convincing, but the problem (as well as “engineering principles” for the smoothing distribution) are probably worth pointing out in the paper itself, since the problem seems likely to occur unless the user is aware of it. (E.g., my proposal of widely separated normals would be a natural distribution to consider until one actually works through the gradients — something not commonly done in the age of autodiff frameworks.)

Another concern is how many sequential operations are needed for inference in the RBM model. (Note: is this actually an RBM, or a general Boltzmann machine?) The q distribution takes the form of an autoregressive model where the variables are processed one at a time. Section 3 mentions the possibility of grouping together variables in the q distribution, and this is elaborated in detail in Appendix A. But the solution requires decomposing the joint into a product of conditionals and applying the CDFs sequentially. So either way, it seems like we’re stuck handling all the variables sequentially, which might get expensive. 

Minor: the second paragraph of Section 3 needs a reference to Appendix A.

[Author Response · Jason Tyler Rolfe · 14 Jan 2017]
**Revised version uploaded**

In response to the helpful comments from the peer reviewers and other readers, I have made some small additions to the paper.  The new version contains:

* A discussion of the relationship to deep belief networks to Section 1.2

* A note in Section 2.1 that a model without continuous smoothing variables zeta can easily be produced by taking the limit beta -> \infty, in which case zeta_i = z_i almost surely

* A discussion of variance of derivatives of the inverse CDF in Appendix D.3

* Some results on simplified probabilistic models in Appendix J

* A few small clarifications.  In particular, I have clarified the use of the term "restricted Boltzmann machine."  The Boltzmann machine used in the prior is bipartite, but all Boltzmann machine variables are directly connected to the smoothing variables zeta, and through them to the rest of the model.

[Final Decision · Program Chairs · 06 Feb 2017]
**ICLR committee final decision**

The authors present a novel reparameterization framework for VAEs with discrete random variables. The idea is to carry out symmetric projections of the approximate posterior and the prior into a continuous space and evaluating the autoencoder term in that space by marginalizing out the discrete variables. They consider the KL divergence between the approximating posterior and the true prior in the original discrete space and show that due to the symmetry of the projection into the continuous space, it does not
 contribute to the KL term. 
 
 One question that warrants further investigation is whether this framework can be extended to GANs and what empirical success they would have.
 
 The reviewers have presented a strong case for the acceptance of the paper and I go with their recommendation.